# Therapeutic targeting of differentiation-state dependent metabolic vulnerabilities in diffuse midline glioma

Nneka E. Mbah[1,2], Amy L. Myers[1,2], Peter Sajjakulnukit [1,3], Chan Chung [1,4,5], Joyce K. Thompson[6], Hanna S. Hong [7], Heather Giza[3], Derek Dang[1,4,8], Zeribe C. Nwosu [2], Mengrou Shan[2], Stefan R. Sweha[1,4,9], Daniella D. Maydan[2], Brandon Chen [2,10], Li Zhang[2], Brian Magnuson[11], Zirui Zhu [12], Megan Radyk[2], Brooke Lavoie[2], Viveka Nand Yadav[13], Imhoi Koo [14], Andrew D. Patterson [15], Daniel R. Wahl [1,16], Luigi Franchi[17], Sameer Agnihotri[18], Carl J. Koschmann[1,17], Sriram Venneti[1,4,17,19] ✉ & Costas A. Lyssiotis [1,2,20,21] ✉

H3K27M diffuse midline gliomas (DMG), including diffuse intrinsic pontine gliomas (DIPG), exhibit cellular heterogeneity comprising less-differentiated oligodendrocyte precursors (OPC)-like stem cells and more differentiated astrocyte (AC)-like cells. Here, we establish in vitro models that recapitulate DMG-OPC-like and AC-like phenotypes and perform transcriptomics, metabolomics, and bioenergetic profiling to identify metabolic programs in the different cellular states. We then define strategies to target metabolic vulnerabilities within specific tumor populations. We show that AC-like cells exhibit a mesenchymal phenotype and are sensitized to ferroptotic cell death. In contrast, OPC-like cells upregulate cholesterol biosynthesis, have diminished mitochondrial oxidative phosphorylation (OXPHOS), and are accordingly more sensitive to statins and OXPHOS inhibitors. Additionally, statins and OXPHOS inhibitors show efficacy and extend survival in preclinical orthotopic models established with stem-like H3K27M DMG cells. Together, this study demonstrates that cellular subtypes within DMGs harbor distinct metabolic vulnerabilities that can be uniquely and selectively targeted for therapeutic gain.

Diffuse midline gliomas (DMG), including diffuse intrinsic pontine gliomas (DIPG), are treatment-resistant and uniformly fatal pediatric brain tumors. The prognosis of this brainstem tumor is dismal with a median overall survival of 9–12 months from diagnosis[1,2]. Radiotherapy is the only therapy that has proven benefit in this patient population. Historically, clinical trials with chemotherapy have failed to demonstrate any additional survival benefit over radiation alone[3,4]. It is therefore imperative to identify strategies for targeting this aggressive and devastating disease.

Recent advances delineating the molecular underpinnings of DMGs and DIPGs revealed that approximately 80% of DIPGs and DMGs harbor a recurrent somatic lysine-to-methionine mutation at position 27 of histone H3.1 (*HIST1H3B*) or H3.3 (*H3F3A*) collectively called H3K27M. H3K27M results in global H3K27 hypo-trimethylation, epigenetic dysregulation, and altered gene transcription[1,5–8]. Other activating mutations and aberrant gene expression patterns identified in DMG include *ACVR1, TP53*, and *ATRX* mutations, and overexpression of transcriptional factors OLIG1 and OLIG2[9–11].

Single-cell RNA sequencing studies from patient tumor samples demonstrate that H3K27M DMGs hijack developmental programs regulating lineage differentiation of neuroglial stem cells[12–14]. Consequently, tumor cells undergo developmental arrest and are locked in a less-differentiated state dependent both on age and anatomic location of the tumor. These studies also revealed that H3K27M DMGs contain a heterogeneous population of cells, where the majority of tumor cells harbor characteristic markers of 'oligodendrocyte precursor cells' (OPC-like), with stem-like and higher renewal potential. The more differentiated-like H3K27M tumor cells exhibit an astrocytic-like (AC-like) phenotype[12]. Of note, in the context of DMG cellular heterogeneity, the H3K27M OPC-like cells are hypothesized to be the putative drivers of tumor growth and aggressiveness and possess in vivo tumor-initiating potential, compared to more differentiated cells[12–18].

Metabolic reprogramming is a hallmark of cancer that influences every aspect of cancer biology[19,20]. Indeed, in comparison to H3 wild-type (H3WT) tumor cells, H3K27M DMG cells enhance metabolic programs including glycolysis, glutaminolysis, and the tricarboxylic acid (TCA) cycle, as well as increase the generation of α-ketoglutarate (α-KG)[21–23]. Of note, enhanced α-KG production in H3K27M DMG was shown to be critical for the maintenance of the preferred global H3K27 hypomethylation status, indicating the important interplay between metabolic reprogramming and epigenetic regulation[24]. More recently, the impiridone ONC201 was shown to be toxic to H3K27M DMG cells by suppressing mitochondrial metabolism and oxidative phosphorylation (OXPHOS)[25,26]. Importantly, recently completed clinical studies with ONC201 show efficacy in H3K27M DMG patients with near doubling in overall survival rates[26–29]. Despite these emerging studies of metabolism in H3K27M DMGs, the role of dysregulated metabolism, particularly in the context of how metabolism impacts stemness and tumorigenicity in H3K27M DMGs, remains largely unknown. Indeed, differential metabolic reprogramming can potentially regulate cancer stemness, differentiation, and cell fate[30]. As a result, many aspects of rewired metabolism can provide therapeutic liabilities within tumors that can be effectively leveraged for therapy.

In this study we seek to elucidate the metabolic dependencies operative in both the stem-like tumorigenic H3K27M gliomas and the more differentiated cell state. By applying a systems biology approach that incorporates metabolomics, transcriptomics, and biochemical analyses, we uncover several nodes of dysregulated metabolic and signaling pathways in the tumorigenic OPC-like versus the more differentiated AC-like DMG populations. This study collectively illustrates that H3K27M DMGs harbor perturbations in metabolic programs that can be exploited for therapeutic benefits.

## Results

### In vitro modeling of the differentiation state of H3K27M DMGs
To study metabolic vulnerabilities associated with the distinctly heterogenous H3K27M DMGs, we generated gliomaspheres (GS) and differentiated glioma cells (DGC) from three patient-derived H3K27M DMG cell lines: DIPG-007, DIPG-XIII and SF7761. It has been previously established that the DMG GS culture conditions (serum-free) enrich for malignant cells that are less-differentiated and stem-like[31,32]. Moreover, DMG GS readily establish tumors following stereotactic injection into the pons of mice[12]. In contrast, culturing the tumor cells in the presence of serum induces differentiation of H3K27M malignant glioma cells (i.e., DGC) and an associated loss or substantially diminished in vivo tumorigenicity[12]. Furthermore, in comparison to the DGC, the GS model most closely recapitulates the phenotype of DMG tumor-xenografts and primary patient tumors[12,31–33].

We maintained these cell lines either as tumorigenic GS cultures (i.e., unattached 3D spheres cultured in serum-free media containing B-27 supplements and the growth factors EGF, FGF and PDGF), or we differentiated the GS cultures into adherent monolayers in the presence of serum (i.e., DGC; Fig. 1A; Supplementary Figs. 1A, 2A). During

the differentiation procedure, we did not observe cell death, arguing against the selection of a rare subclone[34,35]. Further, we performed STR profiling on the paired lines, which indicated that they are genetically identical at the loci assessed (Supplementary Data 1). Evaluation of the growth kinetics of GS vs. DGC revealed that DIPG-007 and SF7661 GS proliferated at a markedly higher rate than their DGC counterparts, indicating that the less differentiated and stem-like GS populations are far more proliferative. The differentiation state did not influence proliferation rates in the DIPG-XIII cells (Fig. 1B).

### Transcriptomic analyzes revealed GS are OPC-like and DGC are AC-like
To characterize the molecular features of the DMG GS and DGC, we performed RNA-sequencing (RNA-seq) to compare the gene expression patterns of GS vs. DGC in these three cell lines. Principal component (PC) analysis of the RNA-seq data revealed that the cell lines clustered based on cell line differences and cellular differentiation status (namely, DGC or GS) (Supplementary Fig. 2B, C). PC1 vs. PC2 was shown to be driven by original cell line differences and clustered based on cell line, irrespective of the differentiation state (Supplementary Fig. 2B). PC3 distinguished between GS and DGC, wherein the cell lines clustered separately, suggesting that PC3 was influenced by the cellular differentiation status. PC3 revealed that GS vs DGC gene expression patterns were more strongly separated in DIPG-007 and SF7761 than in DIPG-XIII (Supplementary Fig. 2C). These observations suggest a potentially marked difference in the resultant phenotype of GS vs. DGC in DIPG-007 and SF7761, which is less pronounced in DIPG-XIII (Supplementary Fig. 1B).

We analyzed genes that were differentially expressed in GS vs. DGC across the three cell lines (DIPG-007, 10276 total genes; SF7761, 9242 total genes; and DIPG-XIII, 8425 total genes) and found 1329 genes and 1163 genes to be commonly upregulated and downregulated, respectively (Fig. 1C, D). The top 50 consistently upregulated and downregulated genes common to the three cell lines revealed distinct gene signatures associated with GS vs. DGC tumor populations. (Supplementary Fig. 2D, E). Next, we examined the gene expression of individual H3K27M DMG stemness and differentiation markers. GS lines upregulated oligodendrocyte transcription factor 2 (OLIG2) (Supplementary Fig. 3A), the gene that encodes a transcription factor typically overexpressed in H3K27M DMGs and critical for the establishment of tumors in vivo[36]. This difference was also observed at the protein level (Fig. 1E). In addition, other H3K27M DMG markers like oligodendrocyte transcription factor 1 (OLIG1) and microtubule associated protein 2 (MAP2) were higher in GS compared to DGC (Supplementary Fig. 3B, C). SRY-box transcription factor 2 (SOX2), epidermal growth factor receptor (EGFR), and Myc proto-oncogene (MYC) are genes whose upregulation is associated with glioma stemness. These genes showed consistently higher expression in DIPG-007 and SF7761 GS compared to their DGC counterparts, and this contrasted with the modestly lower expression in DIPG-XIII GS vs. DGC (Supplementary Fig. 3D–F). Analysis of glial fibrillary acidic protein (GFAP), vimentin (VIM), and S100 calcium binding protein A10 (S100A10), which are typically associated with astrocytic-like differentiated DMG[37–39], revealed substantial upregulation in DGC compared to GS across the three lines (Supplementary Fig. 3G–I).

Malignant H3K27M gliomas have been reported to include tumor cell types that exhibit four main gene signatures: namely, (i) OPC-like cells, (ii) cell cycle (CC), (iii) oligodendrocytes (OL), and (iv) AC-like cells[12]. We cross referenced our bulk RNA-seq gene expression data with that of the published single-cell RNA-seq dataset[12] (GSE102130) that described the four gene signatures. We found that the gene expression pattern observed in GS showed increased enrichment for the OPC-like gene signature, while the gene signature of DGC was consistent with an AC-like phenotype (Fig. 1F). Interestingly, individual gene expression trends observed in our in vitro-generated GS vs. DGC

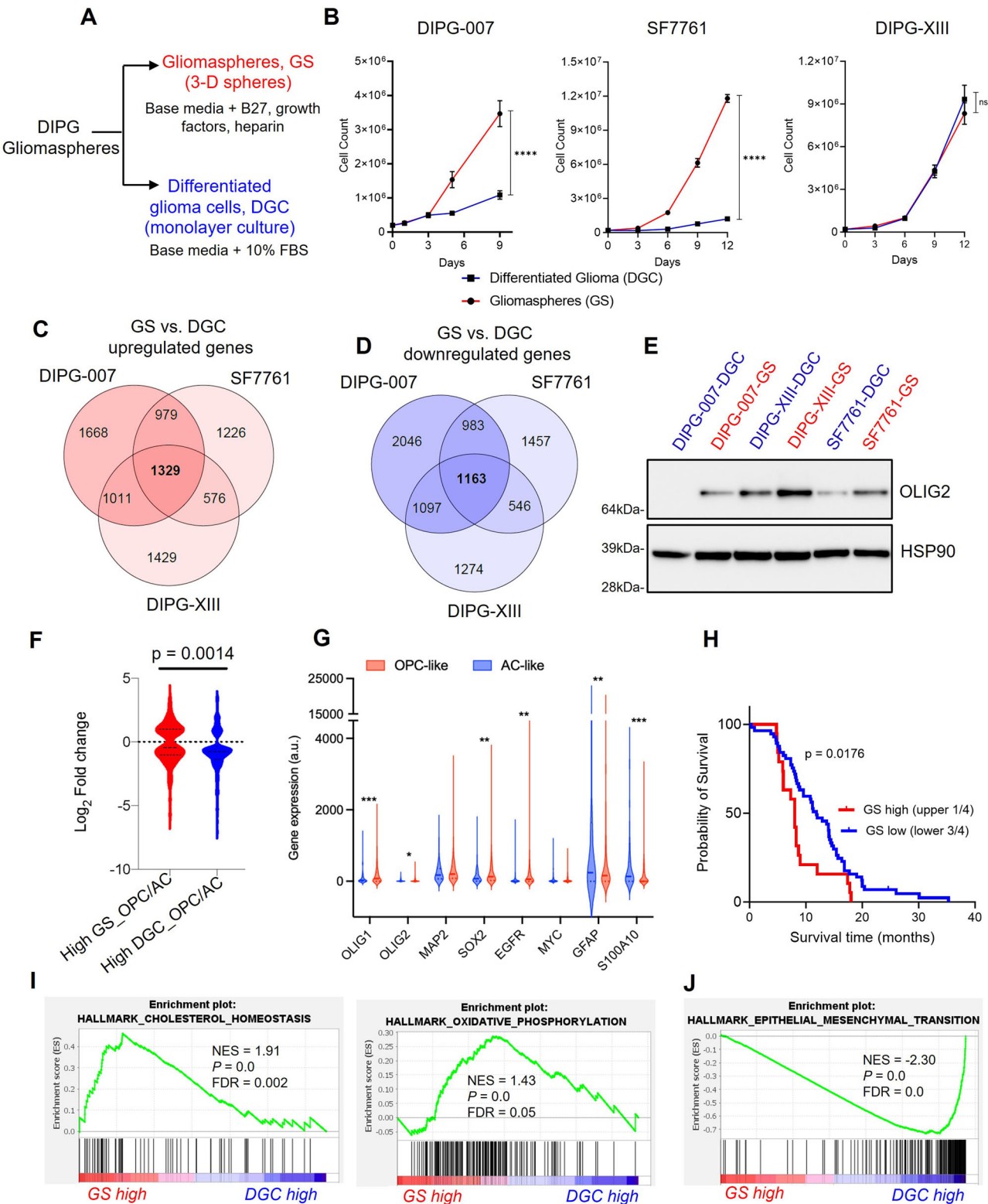

cells (Supplementary Fig. 3A–I) were also recapitulated in the OPC-like vs. AC-like H3K27M cells defined in this published single-cell RNA-seq dataset (Fig. 1G).

Collectively, these data suggest that in vitro generated GS largely represent the OPC-like H3K27M DMG phenotype, which is known to be less-differentiated, stem-like, and exhibit tumor-propagating potential in vivo, while DGC represent the more differentiated AC-like phenotype. Accordingly, the GS and DGC in vitro models developed herein

molecularly mimic two distinct and predominant populations in the heterogenous H3K27M DMG tumor.

### GS gene signature predicts decreased survival of H3K27M DMG patients

To determine the clinical relevance of the GS gene signature in predicting disease outcome and survival of patients with H3K27M DMG, we mined a patient dataset from Mackay et al.[40] containing gene expression

**Fig. 1 | In vitro models of H3K27M DIPG molecularly mimic oligodendrocyte precursor (OPC)-like and differentiated astrocyte (AC)-like DIPG and exhibit distinct gene expression programs. A** Schematic depicting the generation of DIPG gliomaspheres (GS) and differentiated glioma cells (DGC): GS (3-D floating spheres) were cultured in serum-free tumor stem cell media containing growth factors and supplements. Monolayer (adherent) DGC were generated by dissociating GS to single cells and culturing for up to 14 days in media containing 10% fetal bovine serum (FBS). **B** Growth kinetics of DIPG-007, SF7761 and DIPG-XIII GS vs. DGC. Error bars are mean ± SD from three biological replicates; ns (not significant) $p = 0.7589$, ****$p < 0.0001$ by area under proliferation curve test followed by two-tailed Student's t-test. **C, D** Venn diagrams derived using DESeq2 package in R with adjustment for multiple comparisons illustrating the overlap of upregulated (**C**) and downregulated (**D**) genes in GS vs. DGC across three cell lines. **E** Western blot for OLIG2 in GS and DGC pairs of DIPG cell lines. HSP90 was used as a loading control. The figure shows representative blots from two independent experiments. **F** Analysis of the gene expression signature of DIPG GS vs. DGC cross-referenced with patient gene signature capturing OPC-like, AC-like gene signatures. **G** Differential expression of DIPG genes in OPC-like and AC-like tumors from patient single-cell RNAseq. **F, G)** Data was analyzed using unpaired, two-sided, two-tailed, Students t-test with 95% confidence intervals *$p < 0.05$; **$p < 0.01$; ***$p < 0.001$. **H** Survival analysis of DIPG/DMG patients ($n = 78$; female = 41, male = 28, unknown = 9. Age range was 1.2–17.9 years (average 9.8 ± 12 years) based on "GS high" versus "GS low" gene signature. Median survival is 11.9 months versus 8 months; Log Rank $p = 0.0176$. **I, J** "Hallmark" gene set enrichment analysis (GSEA) indicating pathways that are (**I**) upregulated and (**J**) downregulated in DIPG GS vs. DGC. GSEA plots show enrichment scores and include values for normalized enrichment score (NES), nominal $p$-value ($P$), and false discovery rate (FDR) $q$-value. For all panels, red indicates GS; blue, DGC.

and survival data for 76 H3K27M DIPG and DMG patients. We segregated patients into "high GS" vs. "low GS" gene expression categories using unbiased K-means clustering and applied Kaplan-Meier survival analysis to define upper quartile as "GS high" vs. "GS low" tumors. The results revealed that patients with "GS high" tumors showed a significantly decreased survival in comparison to patients in the "GS low" category within H3K27M tumors (Fig. 1H; Supplementary Fig. 3J, K). This result supports our observation that the "GS" gene signature, which recapitulates those of the less differentiated OPC-like cells, represents the more aggressive and tumorigenic cell-state of DMG.

## OPC-like GS upregulate cholesterol metabolism and oxidative phosphorylation

To interrogate the gene expression programs that distinguish the cell state among our cell line pairs, we performed gene set enrichment analysis (GSEA) on each of our RNA-seq datasets from the three lines. Across the three lines, H3K27M DMG GS vs. DGC upregulated genes were associated with the MYC pathway, PI3K/MTORC1 signaling, G2M checkpoint, DNA repair, and E2F signaling (Supplementary Fig. 4A). These pathways have been previously reported to be upregulated in primary patient H3K27M DMG tumors and xenografts[40–42], thereby providing further confidence in our DMG models and analyses. Furthermore, in at least two of the three cell lines, we observed considerable DMG GS enrichment of metabolic pathways, namely cholesterol homeostasis and mitochondrial oxidative phosphorylation (OXPHOS) (Fig. 1I). In contrast, DMG DGC upregulated genes associated with epithelial-mesenchymal transition (EMT) (Fig. 1J), xenobiotic metabolism, inflammatory response, and transforming growth factor beta (TGFβ) signaling (Supplementary Fig. 4B). These findings suggested that tumorigenic OPC-like GS may exhibit enhanced reliance on cholesterol metabolism and mitochondrial OXPHOS programs, which could represent an actionable metabolic vulnerability.

Given the metabolic signatures evident in the transcriptomic profiling, we performed liquid chromatography-coupled mass spectrometry (LC/MS)-based metabolomics[43] to gain a deeper understanding of metabolic differences between the DMG cellular differentiation states. These data revealed that GS and DGC exhibit distinct metabolic landscapes (Supplementary Fig. 5). By taking the average of the three lines, we found that, compared to DGC, GS showed a difference in nucleotide metabolism, lipid and sterol biosynthesis, and amino acid metabolism (Fig. 2A, B). Further, among the ~223 metabolites measured, we found 70 that were consistently altered between GS and DGC. Of these, 45 metabolites were highly increased, and 25 metabolites were decreased in GS compared to DGC (Fig. 2C).

Several metabolites were altered in glycolysis, the TCA cycle, and the purine biosynthesis pathway (Supplementary Fig. 6A). Glycolytic metabolites were generally more abundant in GS compared to DGC (Fig. 2D), with several differences being greater than 10-fold, particularly those in the preparatory phase of glycolysis. Pyruvate and lactate,

products of aerobic and anaerobic glycolysis, respectively, showed either no difference between the differentiation states or were modestly altered (Fig. 2D). Glycolysis connects to the TCA cycle via the generation of acetyl-CoA from CoA and pyruvate. Metabolites in the TCA cycle were generally decreased in DIPG-007 and SF7761 GS compared to DGC, while few differences were observed in the DIPG-XIII cell line pair. An exception was malate, which was increased in GS across the three cell lines (Fig. 2E). The metabolomics studies also revealed increased levels of CoA and carnitine, key metabolites and rate-limiting substrates in lipid and sterol biosynthetic pathways (Supplementary Fig. 6B).

Despite the marked differences in glycolytic and TCA cycle intermediates, consistent differences were not observed for expression of glucose or glutamine transporters (Supplementary Fig. 7A–C). Additionally, we found that DIPG-007 GS cultures are more sensitive to glucose deprivation than DGC, though this did not have an impact on cellular phenotype (Supplementary Fig. 7D, E). Glutamine deprivation was well-tolerated by both GS and DGC cultures, again without having an impact on cellular phenotype (Supplementary Fig. 7D, F).

## Purine nucleotides are enriched in OPC-like GS

Of the phosphorylated purine species detected, markedly higher levels of purine nucleotide pools (>50-fold in several cases) were observed in GS compared to DGC in DIPG-007 and SF7761, including adenosine monophosphate (AMP), guanosine monophosphate (GMP), inosine monophosphate (IMP), inosine diphosphate (IDP) and adenosine diphosphate (ADP) (Fig. 2F). Apart from ADP, such differences were not observed in DIPG-XIII (Fig. 2F). Increased expression of genes encoding purine pathway enzymes were similarly observed (Supplementary Fig. 6C), including phosphoribosyl pyrophosphate synthetase 2 (*PRPS2),* which converts ribose-5-phosphate to phosphoribosyl pyrophosphate; adenylosuccinate synthase 2 (*ADSS*), which converts IMP to adenylosuccinate; adenylosuccinate lyase (*ADSL*), which converts adenylosuccinate to AMP; inosine monophosphate dehydrogenase 1 (*IMPDH1*), which converts IMP to xanthine monophosphate (XMP); and guanine monophosphate synthase (*GMPS*), which converts XMP to GMP (Supplementary Fig. 6A). Indeed, increased purine nucleotides pools have been demonstrated to be an intrinsic characteristic of brain tumor-initiating glioma cells[44].

## AC-like DGC accumulate metabolites associated with cellular differentiation

Comparison of metabolite abundance in DGC relative to GS revealed increases on the order of 10-fold for taurine, creatine, creatinine, uric acid, and hydroxyproline (Fig. 2G). These data were provocative because taurine, creatine, and creatinine have been shown to be elevated in oligodendrocytes generated by inducing differentiation of primary OPCs using triiodothyronine (T3)[45]. Moreover, exogenous taurine was shown to promote drug-induced differentiation of primary

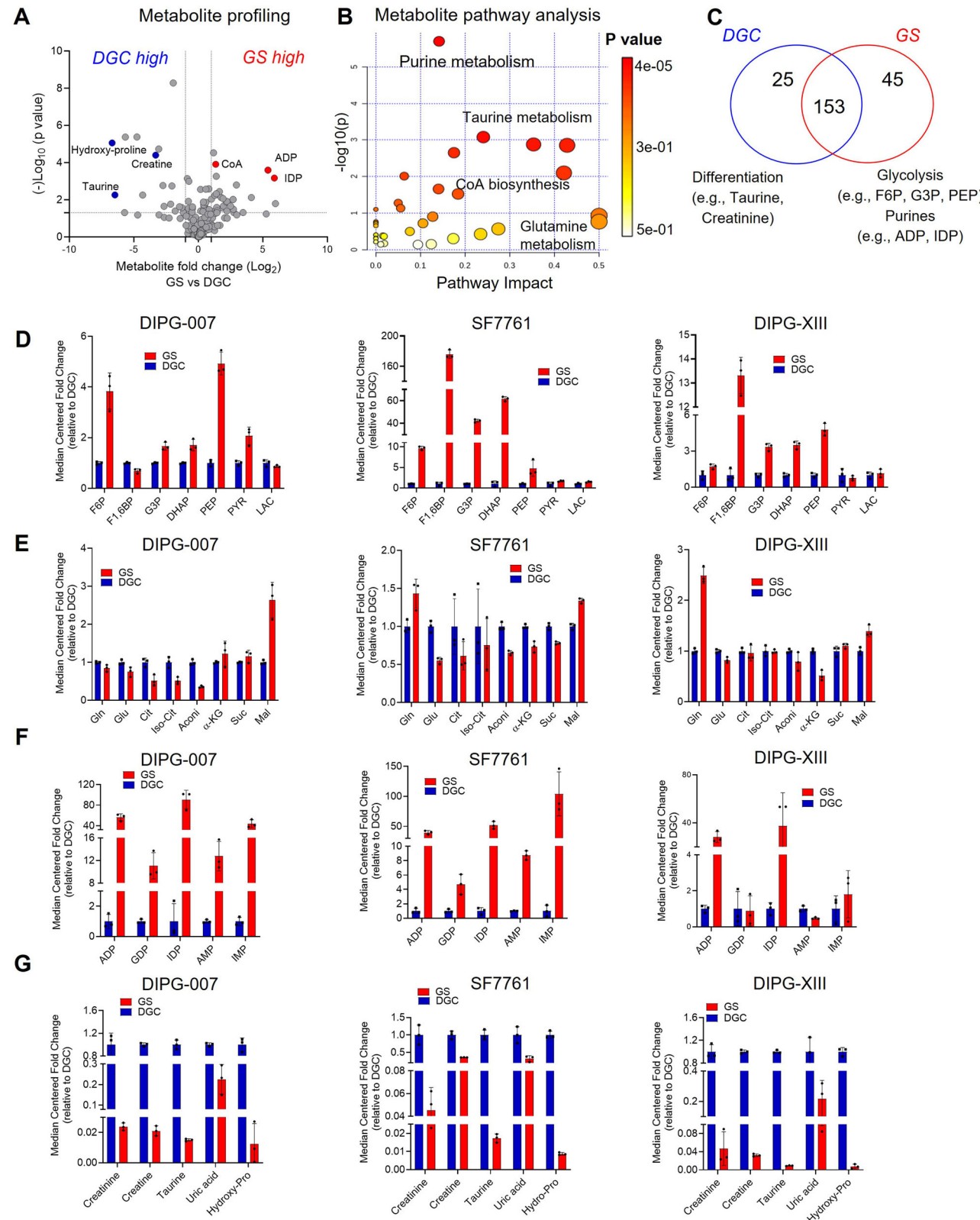

OPC cells to OLs and is presumed to be synthesized by cells to promote lineage differentiation[45].

## AC-like DGC upregulate EMT pathway genes and are vulnerable to ferroptosis

The results from our transcriptome and metabolome profiling efforts identified several nodes of metabolism that differ between GS and DGC. We next sought to assess if these differences in metabolic programming provide therapeutic vulnerabilities. GSEA illustrated that DGC upregulated genes associated with epithelial-mesenchymal transition (EMT) and TGFβ signaling (Fig. 1J; Supplementary Fig. 4B). This EMT signature was characterized by a general increase in expression of EMT marker genes, including transforming growth factor beta 2 (*TGFB2*), vascular cell adhesion molecule 1 (*VCAM1*), snail family

**Fig. 2 | Metabolomic profiling of gliomaspheres (GS) and differentiated glioma cells (DGC) reveals differentiation state-dependent metabolic features.**
**A** Volcano plot indicating differential metabolite profiles in GS vs. DGC, presented as the average metabolite abundance value from the three cell lines DIPG-007, SF7761, and DIPG-XIII. **B** Metabolic pathway enrichment as determined using MetaboAnalyst based on differential metabolite abundance in GS vs. DGC. **A**, **B** Statistically significant differential metabolites evaluated using t-test with two-tailed distribution and two-sample unequal variance based on p-values from 3 independently prepared samples. Dot size reflects pathway impact. **C** Venn diagram indicating the number of metabolites significantly altered in GS vs. DGC across all three cell lines. Example metabolites and pathways are indicated below.
**D–F** Differential abundance of select metabolites for (**D**) glycolysis, (**E**) tricarboxylic acid (TCA) cycle, and (**F**) purine nucleotides in DIPG-007, SF7761, and DIPG-XIII GS

versus DGC. **G** Highly enriched metabolites in DIPG DGC vs. GS. **D–G** Metabolite levels presented as median-centered fold change of GS relative to DGC across the three cell lines. Error bars represent mean ± SD. All metabolomic profiling data (panels **A–G**) were generated from the average of three independently prepared samples run on the same day. Source data are provided as Source data file 2. F6P, fructose-6-phosphate; F1,6BP, fructose-1-6-bisphosphate; G3P, glyceraldehyde-3-phosphate; DHAP, dihydroxyacetone phosphate; PEP, phosphoenolpyruvate, PYR, pyruvate; LAC, lactate; Gln, glutamine; Glu, glutamate; Cit, citrate; Iso-Cit, iso-citrate; Aconi, cis-aconitate; α-KG, alpha-ketoglutarate; Suc, succinate; Mal, malate; ADP, adenosine diphosphate; GDP, guanosine diphosphate; AMP, adenosine monophosphate; IMP, inosine monophosphate; IDP, inosine diphosphate; Hydro-Pro, hydroxyproline. For panels **A**, **C–G**, red indicates GS; blue, DGC.

transcriptional repressor 1 (*SNAI1*), matrix metallopeptidase 2 (*MMP2*), and matrix metallopeptidase 11 (*MMP11*) (Fig. 3A). Several recent studies have illustrated that the mesenchymal state of cancer cells exposes a vulnerability to ferroptosis, a form of metabolic-stress cell death induced by inhibiting the GPX4 lipid peroxidase pathway[46,47]. Ferroptosis can be induced by genetic or pharmacological manipulations that impair cystine uptake, block glutathione (GSH) synthesis, or directly inhibit activity of the central lipid peroxidase, GPX4[48] (Fig. 3B). Based on this knowledge, we hypothesized that DGC, owing to its high mesenchymal gene signature, would be susceptible to GPX4 inhibition and ferroptosis.

Treatment of DIPG-007, SF7761, and DIPG-XIII DGC with the GPX4 inhibitor RSL3 led to profound cell death by 48 h (Fig. 3C). To determine whether the RSL3-induced cell death was indeed ferroptotic in nature, we demonstrated that pre-treating cells with the lipophilic antioxidant ferrostatin-1 (Fer-1; a well-established inhibitor of ferroptosis[49]) rescued cell death. In contrast to DGC, GS cells exhibited RSL3-induced cytotoxicity at much higher concentrations, and more importantly, this effect could not be rescued substantially by Fer-1. The lack of rescue illustrates a ferroptosis-independent mechanism of cell death (Fig. 3D).

Next, we assessed if RSL3 could induce lipid oxidation, a classic hallmark of ferroptosis, in DGC. Indeed, treatment of DIPG-007 with RSL3 resulted in increased accumulation of lipid reactive oxygen species (lipid ROS), as measured by C-11 BODIPY, which could be mitigated by pre-treating cells with Fer-1 (Fig. 3E). To rule out other avenues of RSL3-induced cytotoxic cell death, DGC were pre-treated with antioxidants and ferroptosis inhibitors (trolox, Fer-1), z-vad-fmk (apoptosis inhibitor), bafilomycin-A1 (autophagic cell death inhibitor), or necrosulfonamide (necroptosis inhibitor). Only the antioxidants rescued cell death induced by RSL3 in DIPG-007 DGC (Fig. 3F). Collectively, these data indicate a ferroptotic-specific mechanism of cell death in DGC.

As an important control, we also investigated whether the resistance of GS to ferroptosis was the result of culture media composition. To test this, we cultured freshly dissociated GS in serum-containing DGC media or GS media with an antioxidant-free B-27 supplement and assessed RSL3-induced ferroptosis. In both instances, GS displayed resistance to ferroptosis regardless of the media formulation (Fig. 3G; Supplementary Fig. 8A–C). These results illustrate that the growth medium itself does not directly impact the susceptibility of DMG cells to ferroptosis. Rather, our data suggest that AC-like DGC have undergone the process of differentiation and, thereby, harbor cell-intrinsic qualities that promote sensitivity to ferroptosis. This is consistent with our hypothesis that the mesenchymal state of DGC sensitizes them to GPX4 inhibition-induced ferroptosis. Of note, analysis of a panel of canonical ferroptosis regulators did not reveal differential expression between GS and DGC cells (Supplementary Fig. 8D), and prolonged low-dose RSL3 treatment did not have an impact on GS cellular phenotype (Supplementary Fig. 8E). Additionally, SF7761 GS cells are highly dependent on antioxidants for survival.

## Cholesterol biosynthesis and mitochondrial OXPHOS are metabolic vulnerabilities in OPC-like GS

Cholesterol homeostasis and mitochondrial OXPHOS were the top upregulated metabolic pathways by gene expression analysis in OPC-like GS (Fig. 1I). Accordingly, we investigated the sensitivity of GS vs. DGC to OXPHOS inhibitors (Phenformin, Metformin, IACS-010759) or cholesterol biosynthesis inhibitors (statins). In vitro cultured DIPG-007 GS and DGC treated with increasing drug concentrations revealed that GS were strikingly and selectively more sensitive to Metformin, Phenformin, and IACS-010759 (Fig. 4A–C), as well as to statins (Atorvastatin, Fluvastatin, and Pitavastatin) (Fig. 4D–F). Of the five clinically available lipophilic statins, Pitavastatin, was most potent at reducing viability of DIPG-007 in vitro (Supplementary Fig. 9A). Similar observations were made in SF7761 (Supplementary Fig. 9B–F) and DIPG-XIII cells, with the notable exception that metformin and phenformin did not exhibit a differential response in DIPG-XIII (Supplementary Fig. 9G-J). Prolonged low-dose Phenformin or Pitavastatin treatment did not have an impact on GS cellular phenotype (Supplementary Fig. 8F). Lastly, we established the PPK murine DMG cell line (P53; PDGFRA; H3K27M), which has been demonstrated to replicate the H3K27M DMG biology[50], as either GS or DGC. GS cultures of PPK exhibited increased sensitivity to Phenformin and Pitavastatin, relative to DGC (Supplementary Fig. 9K, L).

To determine the mechanism of cytotoxicity induced by OXPHOS inhibition (Phenformin) and statins (Pitavastatin), we treated DIPG-007 DGC and GS with equal concentrations of these compounds and assessed PARP cleavage via western blot as a readout for apoptotic cell death. The results showed modestly elevated levels of PARP cleavage in DIPG-007 GS but not in DGC, demonstrating that the cytotoxic effects of statins and OXPHOS inhibition are at least partially the result of induction of apoptotic cell death (Fig. 4G, H).

In an attempt to invoke an even more potent cytotoxic effect using both pathway inhibitors, DIPG-007 GS and murine PPK GS were treated with increasing doses of single agent Pitavastatin, Phenformin, or the combination with the IC$_{25}$ or IC$_{50}$ of the respective combinatorial compound. The results revealed sub-additive activity in DIPG-007 (Supplementary Fig. 9M, N). No additional cytotoxic benefit was found in murine PPK GS treated with either Metformin or Phenformin in combination with Pitavastatin (Supplementary Fig. 9O–R).

## OPC-like GS depend on the sterol biosynthesis pathway for cholesterol

Based on the upregulation of cholesterol metabolism gene expression (Fig. 1I, Supplementary Fig. 10A–C) and the robust sensitivity of GS to statins, we hypothesized that GS are metabolically dependent on an output of the sterol biosynthetic pathway for survival.

To test this hypothesis, we grew DIPG-007 GS and DGC cells in media supplemented with isotopically labeled glucose (uniformly carbon-13; U13C-glucose), and collected samples at 15 min, 1 h, 4 h, and 24 h. These were analyzed by LC/MS-based metabolomics to measure the flux of glucose carbon entry into glycolysis, the TCA cycle, TCA

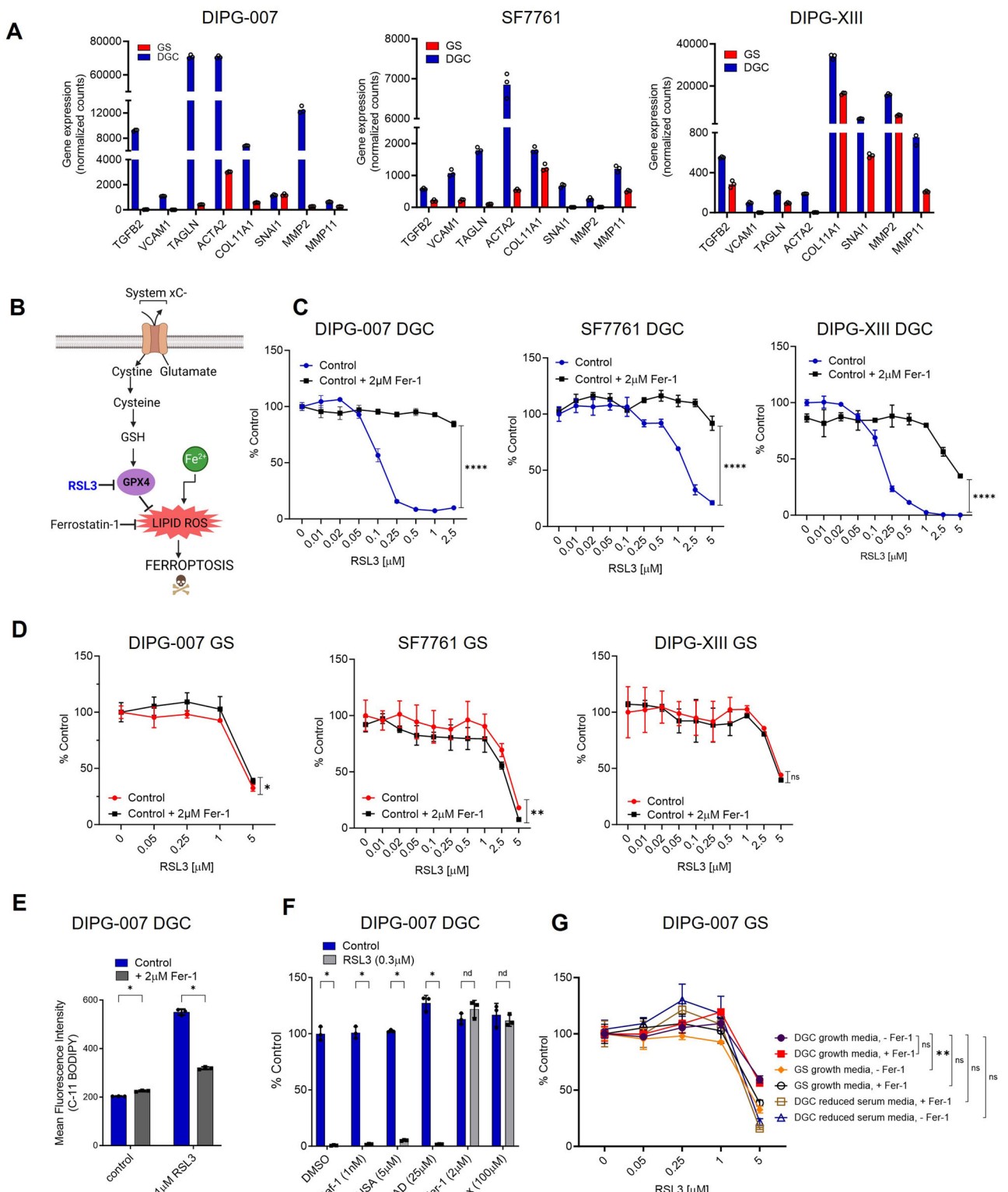

cycling, de novo lipid biosynthesis, and the sterol biosynthetic pathway. We found that glucose entry into glycolysis and the upper TCA cycle (i.e. citrate) was comparable between DGC and GS (Fig. 5A, C; Supplementary Fig. 10D–G; Supplementary Fig. 11A, B). TCA cycling and anaplerosis were reduced in GS, compared to DGC (Fig. 5B, D; Supplementary Fig. 10H–K). Glucose carbon entry into the lower TCA cycle was markedly slower in GS, potentially because citrate was being siphoned to support sterol biosynthesis. Indeed, we observed that sterol biosynthesis was faster in GS, as measured by labeling of the

sterol biosynthesis intermediate mevalonate (Fig. 5E). Further, GS had increased rates of de novo lipid biosynthesis (Fig. 5F; Supplementary Fig. 11C–I). In sum, these observations reflect a higher demand for GS cells on endogenously produced cholesterol, whose inhibition presents a metabolic vulnerability.

Statins inhibit HMG-CoA reductase (HMGCR), the first step in the sterol biosynthesis pathway, whose outputs include cholesterol, protein post-translational modifications (e.g., farnesyl, geranyl), steroid hormones, and coenzyme Q10 (CoQ10)[51,52] (Fig. 5G). To determine the

**Fig. 3 | Ferroptosis is a metabolic vulnerability of differentiated glioma cells (DGC). A** Differential expression of epithelial-mesenchymal transition (EMT) genes by bulk transcriptomics in DIPG-007, SF7761, and DIPG-XIII gliomaspheres (GS) vs. DGC. Data presented from three biological replicates. **B** Simplified scheme depicting the role of glutathione peroxidase 4 (GPX4) in ferroptosis. **C, D** RSL3 dose-response in (**C**) DGC and (**D**) GS DIPG-007, SF7761, and DIPG-XIII with or without 1 h ferrostatin-1 (Fer-1) pre-treatment. **E** Flow cytometry assessment of intracellular lipid reactive oxygen species (ROS) using C-11 BODIPY in DIPG-007 DGC treated with RSL3 for 6 h with or without 1 h Fer-1 pre-treatment. Data expressed as mean ± SD mean fluorescent intensity (MFI) (****$p < 0.0001$; multiple unpaired t test). **F** Cell viability of RSL3-treated DIPG-007 DGC in the presence of bafilomycin A1 (Baf-1), necrosulfonamide (NSA), ZVAD-FMK (Z-VAD), Fer-1, or Trolox. **G** RSL3 dose response in DIPG-007 GS cultured in antioxidant-free B-27-supplemented GS, DGC, or reduced-serum (2.5%) DGC media, with or without 1 h Fer-1 pre-treatment. Cell viability (**C, D, G**) assessed with Cell Titer-Glo 2.0 (DGC) or Cell Titer-Glo 3D (GS) at 48 h and data expressed as percent vehicle control (0.1% DMSO). Error bars represent mean ± SD from three biological replicates with ns = not significant, *$p < 0.05$, **$p < 0.01$, ****$p < 0.0001$ by area under proliferation curve test followed by either two-tailed Student's t-test) (**C, D**) or one-way ANOVA for multiple Uncorrected Fisher's LSD test (**G**). For panels **E** and **F**, error bars represent mean ± SD from three biological replicates with nd = no discovery, *$p < 0.00001$ by multiple unpaired t-test and Two-stage step-up. Panels **C**–**G** are representative of data from three biological replicates. TGFB2, transforming growth factor beta-2; VCAM1, vascular cell adhesion molecule 1; TAGLN, transgelin; ACTA2, actin alpha 2; COL11A1, collagen type XI alpha 1 chain; SNAI1, snail family transcriptional repressor 1; MMP2, matrix metallopeptidase 2; MMP11, matrix metallopeptidase 11; GSH, glutathione. For panels A and C-F, red indicates GS; blue, DGC. Figure 3B created with BioRender.com released under a Creative Commons Attribution-NonCommercial-NoDerivs 4.0 International license.

arm of the sterol biosynthesis pathway involved in mediating GS sensitivity to statins, we treated DIPG-007 GS in vitro with Pitavastatin alone or in combination with key intermediates of the sterol biosynthesis pathway, including mevalonate, farnesyl pyrophosphate (FPP), or geranylgeranyl pyrophosphate (GGPP) (Fig. 5G). The results revealed that mevalonate, a rate limiting metabolite in the sterol biosynthesis pathway, protected cells from effects of Pitavastatin (Fig. 5H). In addition, FPP and GGPP partially rescued Pitavastatin-induced loss of cell viability (Fig. 5I, J).

We next investigated whether addition of exogenous cholesterol, an end product of the pathway, could similarly protect GS cells from Pitavastatin-induced cytotoxicity. To deliver cholesterol, we used cholesterol conjugated to methyl-β-cyclodextrin to promote cell permeability. This was added to cells in combination with Pitavastatin. Here, we observed that cholesterol robustly rescued the loss of viability induced by Pitavastatin, indicating a dependency of GS on cholesterol for survival (Fig. 5K). Similar observations were made using SF7761 and DIPG-XIII GS (Supplementary Fig. 12A–F). CoQ$_{10}$ acts as an electron shuttle between complexes II and III of the electron transport chain and is, thus, an important mediator of OXPHOS. However, CoQ$_{10}$ media supplementation did not protect cells to the same extent as cholesterol or mevalonate. These results suggest that the cytotoxic effect of Pitavastatin is not the result of indirect inhibition of mitochondrial respiration via limiting CoQ$_{10}$ biosynthesis (Fig. 5L; Supplementary Fig. 12G, H).

## OPC-like GS exhibit decreased bioenergetic capacity and activity

To gain insights on why mitochondrial OXPHOS is a metabolic dependency in DMG GS, we evaluated the bioenergetic capacity of untreated GS compared to DGC using the Seahorse extracellular flux analyzer. We monitored the oxygen consumption rate (OCR), which is an indicator of mitochondrial respiration. The results showed that the basal OCR was lower in GS compared to DGC in DIPG-007 and SF7761, with a trend toward lower OXPHOS in DIPG-XIII GS (Fig. 6A, B). Further, challenge with the ATP synthase inhibitor, oligomycin, decreased respiration move severely in GS. And, most strikingly, treatment with the mitochondrial membrane potential uncoupler FCCP, which facilitates maximal oxygen consumption in the mitochondria, revealed that GS displayed decreased OCR compared to DGCs (Fig. 6A). The modest decrease in basal OCR and the OCR response to Oligomycin or FCCP in DIPG-XIII GS vs. DGC (Fig. 6A, B) is consistent with the lack of differential sensitivity to metformin and phenformin (Supplementary Fig. 9G, H). Next, we assessed spare respiratory capacity (SRC), a measure of the difference between maximal oxygen consumption capacity and basal oxygen consumption in the mitochondria. SRC was similarly reduced in GS vs. DGC (Fig. 6C). In alignment with our findings, cells with low SRC have been reported to be relatively proliferative, and low SRC is associated with stem-like cells, while SRC is elevated in differentiated cells[53,54] (Fig. 1B).

The seeming discrepancy between the upregulated mitochondrial OXPHOS gene signature (Fig. 1I), decreased TCA cycling (Fig. 5B–D; Supplementary Fig. 10H–K), and the decreased OCR and SRC parameters in GS populations (Fig. 6B, C) motivated us to take a more detailed look at the bioenergetic charge in our cultures. To this end, we interrogated our in-house metabolomics profiling dataset and determined the NAD/NADH ratios, the adenylate energy charge, and the ATP/ADP ratio of the cells. First, differences were not observed in whole cell NAD/NADH ratios across pairs of GS and DGC culture models (Supplementary Fig. 12I). The adenylate energy charge (AEC) of a cell is an index of the energetic status of the cell that considers the differential intracellular levels of the adenylate pool, namely adenosine triphosphate (ATP), adenosine diphosphate (ADP), and adenosine monophosphate (AMP)[55,56]. The AEC is calculated by applying the formula $[(ATP) + 0.5(ADP)]/[(ATP) + (ADP) + (AMP)]$, which yields values between 0 and 1 wherein normal cells remain in the 0.7 to 0.95 range[55,56]. Assessment of the AEC in GS vs. DGC lines revealed that tumorigenic DIPG-007 and SF7761 GS displayed lower AEC values compared to their DGC counterparts (Fig. 6D). This result is indicative of a consequent greater dependency of GS on ATP-generating pathway(s), chief among which is mitochondrial OXPHOS. Here again, modest differences in the AEC values were observed for DIPG-XIII. This result is consistent with our observation that DIPG-XIII GS do not demonstrate the same degree of differential and selective sensitivity to OXPHOS inhibition (Supplementary Fig. 9G, H). Despite the variability in AEC among the DMG pairs, the direct ratio of ATP to ADP revealed lower levels in GS across all lines (Fig. 6E). Thus, the low energy charge of GS indicates a DMG metabolic state where catabolic processes to regenerate ATP are limiting, which we put forth provides the explanation for the therapeutic susceptibility to OXPHOS inhibition.

Thus, we next analyzed glycolytic flux by measuring extracellular acidification rate (ECAR) using the Seahorse bioanalyzer. While upstream glycolytic pools were greatly enriched in DIPG-007 and SF7761 GS by metabolomics analysis (Fig. 2D), ECAR was more pronounced in their respective DGC counterparts (Fig. 6F). These results suggest that DIPG-007 and SF7761 DGC can compensate for the inhibition of respiration through utilization of glycolysis, which GS appear unable to do, potentially because of glycolytic stalling as reflected in the large metabolite pool sizes.

In summary, these results reveal that OXPHOS inhibitor-sensitive GS have lower OCR, SRC, AEC, and ECAR. This suggests that DIPG-007 and SF7761 GS exist in a lower and more vulnerable bioenergetic state than their DGC counterparts, providing important insight into why OPC-like GS are highly sensitive to mitochondrial targeting.

## OPC-like GS are sensitized to radiotherapy

Mitochondrial SRC correlates with the capacity of cells to respond or adapt to stress conditions (e.g. oxidative stress)[53]. We therefore

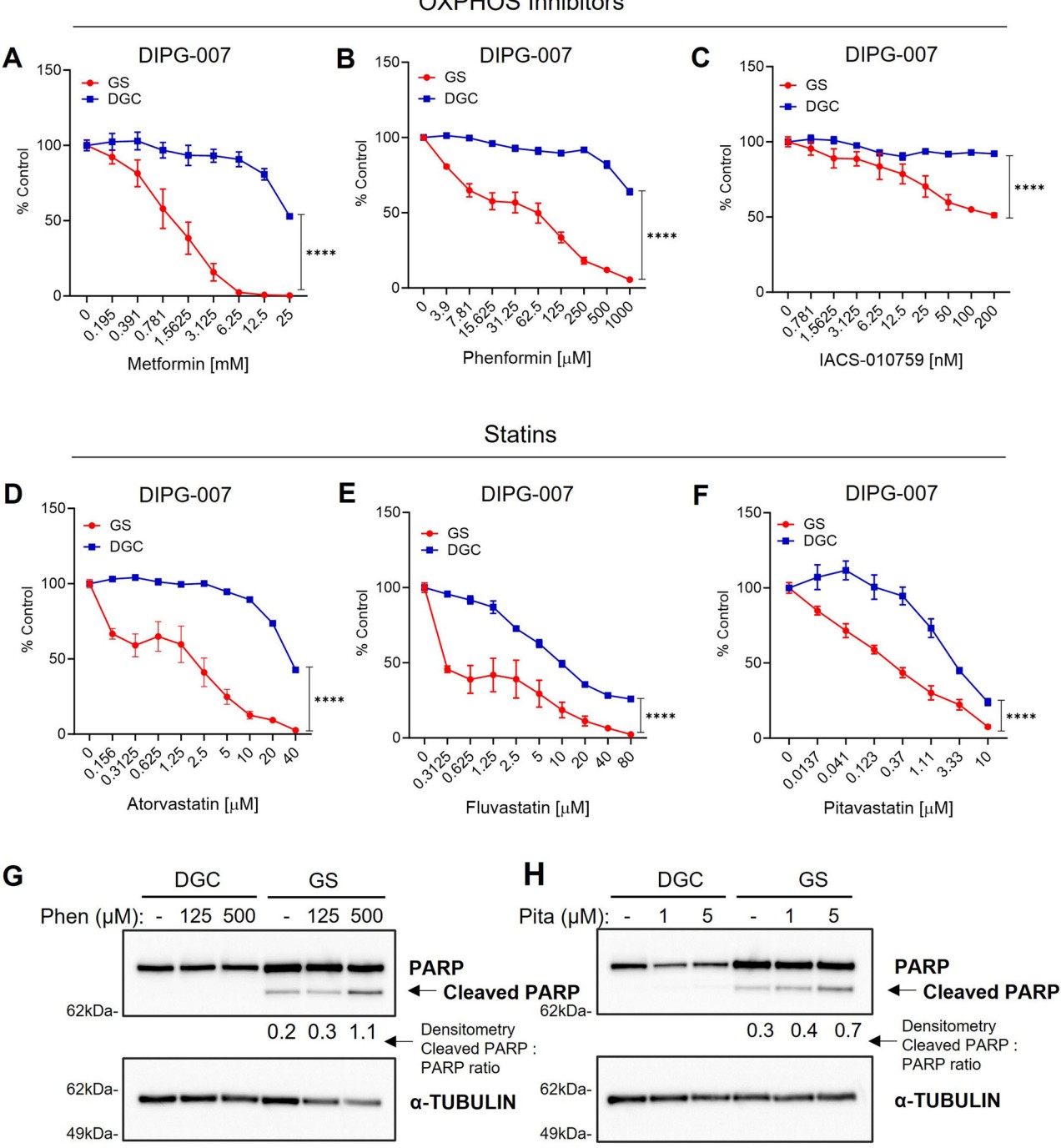

**Fig. 4 | Oxidative phosphorylation (OXPHOS) and cholesterol biosynthesis are targetable vulnerabilities in oligodendrocyte precursor (OPC)-like gliomaspheres (GS).** A–F Dose-response curves for DIPG-007 GS vs. differentiated glioma cells (DGC) treated with indicated concentrations of mitochondrial OXPHOS inhibitors (i.e. complex I inhibitors) (**A**) Metformin, (**B**) Phenformin, and (**C**) IACS-010759 (IACS), or statins (**D**) Atorvastatin, (**E**) Fluvastatin, and (**F**) Pitavastatin for 3 days or 7 days (Metformin). Cell viability was assayed using Cell Titer-Glo 2.0 (DGC) or 3D (GS) and results expressed as percent of vehicle control (0.1% DMSO). Error bars for panels **A–F** represent mean ± SD from three biological replicates (****$p < 0.0001$ by area under proliferation curve test followed by two-tailed Student's t-test). **G, H** Western blot analysis of PARP cleavage (apoptosis indicator) with an α-TUBULIN loading control in (**G**) Phenformin (Phen)-treated and (**H**) Pitavastatin (Pita)-treated DIPG007 DGC and GS. The figure shows representative blots from two independent experiments. Cells were treated for 48 h at the indicated concentrations. For panels **A–F**, red indicates GS; blue, DGC.

hypothesized that lower SRC in the GS would be reflected in an increased susceptibility to ionizing radiation, the mainstay therapy for DMG and a well-established inducer of cytotoxic oxidative stress. To this end, we treated the DMG cells with varying doses of radiation and evaluated cell viability after 7 days. With the exception of DIPG-XIII, the GS were markedly more sensitive to radiation treatment than DGC

(Fig. 7A). Additionally, we examined the combination of radiation and statins or OXPHOS inhibitors. DIPG-007 GS cultures were treated with a dose response of atorvastatin or phenformin plus or minus 2 Gy radiation. At some combinations, modest additive cell killing was observed. However, synergy was not observed, and the additivity window was narrow (Supplementary Fig. 11J, K).

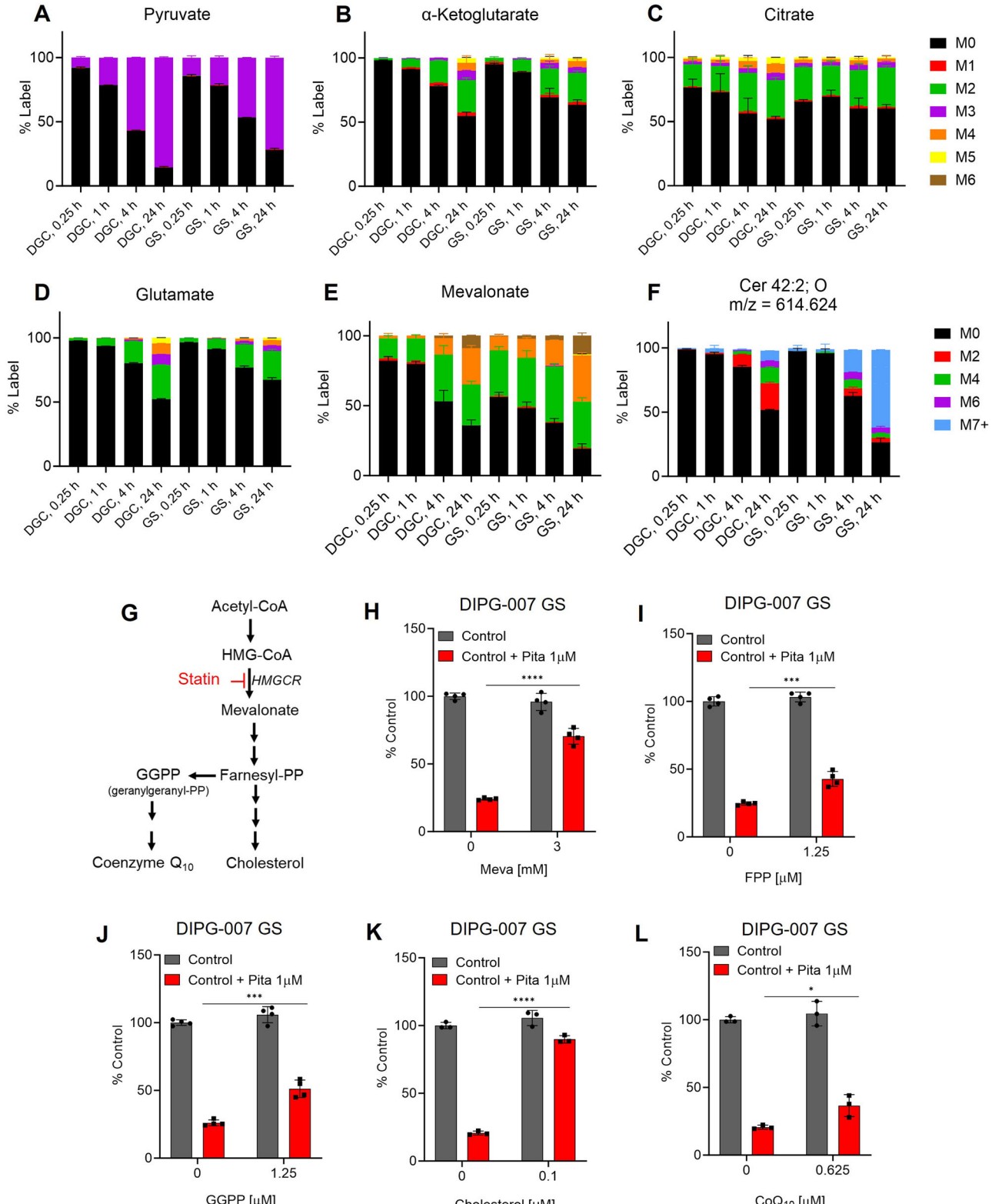

## Cholesterol biosynthesis and OXPHOS inhibition decrease tumor burden and increase overall survival of DMG tumor bearing mice

Statins are used to lower cholesterol and protect from cardiovascular disease and represent one of the most widely used drugs in the clinic, illustrating their safety and tolerability[57,58]. Similarly, biguanides, which act through OXPHOS inhibition[59,60], are clinically deployed to reduce blood glucose in diabetes and have seen recent application in cancer

trials, again illustrating the potential for rapid deployment in clinical trials for DMG. Furthermore, studies have shown that biguanides can modestly transverse the blood brain barrier (BBB)[61,62], and some classes of statins display brain penetrance, depending on the pharmacophore, including Pitavastatin[63,64].

Thus, to evaluate the effects of OXPHOS inhibitors and statins on tumor growth and overall survival, we employed a preclinical orthotopic mouse model of DIPG in which bioluminescent DIPG-007 cells,

**Fig. 5 | Oligodendrocyte precursor (OPC)-like gliomaspheres (GS) exhibit enhanced sterol biosynthetic activity to make cholesterol. A–F** Metabolomics-based assessment of glucose-derived carbon entry into downstream metabolism in differentiated glioma cells (DGC) or GS at the indicated timepoints. Representative metabolites are presented for (**A**) glycolysis, (**B–D**) the TCA cycle, (**E**) sterol biosynthesis, and (**F**) de novo fatty acid biosynthesis. Enrichment is presented as mass + the number of carbons labeled in the metabolite, m + n. Data are presented as the fractional enrichment of the pool. Colored, stacked bars represent the isotopologues, and the isotopologue enrichment within each group is determined by the mean calculated from three biological replicates with ± SD error bars indicated. Cer (XX:X;OX), ceramide (carbons:unsaturation;oxygens); m/z, mass/charge.

(**G**) Schematic of sterol biosynthesis pathway indicating key intermediates in the biosynthesis of cholesterol and coenzyme $Q_{10}$. **H–L** Cell viability of DIPG-007 GS following treatment with vehicle (0.4% DMSO) or Pitavastatin (Pita) with or without co-treatment with (**H**) mevalonate (Meva), (**I**) farnesyl pyrophosphate (FPP), (**J**) geranylgeranyl pyrophosphate (GGPP), (**K**) cholesterol, and (**L**) coenzyme $Q_{10}$ ($CoQ_{10}$) at the indicated concentrations. Cell viability was assayed at 72 h post-treatment using Cell Titer-Glo 2.0. Results expressed as percent of vehicle control and error bars represent mean ± SD from three biological replicates (ns = not significant; *$p < 0.05$; **$p < 0.01$; ***$p < 0.001$; ****$p < 0.0001$ by Unpaired t test, two-tailed with 95% confidence interval). HMG-CoA, β-Hydroxy β-methylglutaryl-coenzyme A; HMGCR, HMG-CoA reductase.

grown under GS conditions, were stereotactically injected into the pons of immunodeficient mice. Cognate DGC cells similarly injected do not form discernible tumors (Supplementary Fig. 13A, B). Tumor engraftment was confirmed via bioluminescent imaging (BLI) 3-weeks post tumor implantation and mice were randomized into four arms receiving vehicle, Pitavastatin (10 mg/kg), Phenformin (50 mg/kg) or a combination of both drugs, administered intraperitoneally (Fig. 7B). These treatment doses were determined from an in-house dose-escalating tolerability study in which no signs of toxicity or weight loss were observed following administration of the drugs over a 2-week course (Supplementary Fig. 13C). Notably, treatment with Pitavastatin or Phenformin resulted in either significant or a trend towards reduction in tumor volume, respectively, based on BLI, with the combination of both drugs not showing improvement over the single agents alone (Fig. 7C). Neither treatment adversely impacted mouse body weight (Fig. 7D). In addition, treatment with Pitavastatin or Phenformin significantly extended the survival of DIPG-007 tumor-bearing mice, and here again, the combination did not provide additional benefit (Fig. 7E), consistent with our in vitro findings (Supplementary Fig. 9M, N). At endpoint, more than one month after treatment commenced, histology indicated that there were no detectable differences in differentiation (i.e. Olig2, GFAP, S100), proliferation (Ki67), or apoptosis (cleaved caspase 3; CC3) among the four study arms (Supplementary Fig. 14). Collectively, single agent metabolic inhibitors showed promising results in providing survival benefits in this preclinical model of DMG, demonstrating the potential utility of targeting cholesterol biogenesis and mitochondrial respiration in DMG patients.

## Discussion

H3K27M DMGs are characterized by intratumoral heterogeneity comprising distinct tumor cell types, wherein the stem-like and tumor-initiating characteristics are driven by a population of less-differentiated OPC-like glioma cells while the more differentiated AC-like glioma cells represent a minority[12]. We demonstrated that this tumor heterogeneity can be modeled in vitro and is substantially recapitulated in DMG GS and DGC, which are enriched for OPC-like and AC-like gene signatures, respectively.

By applying a systems biology-driven approach that encompassed transcriptomics, metabolomics, and bioenergetic analysis, we showed that the OPC-like and AC-like tumor phenotypes harbor distinct metabolic vulnerabilities. Compared to DGC, GS populations showed higher levels of purine nucleotides. This finding is consistent with features of stem-like brain tumor-initiating cells described in adult glioblastoma, which upregulate purine synthetic intermediates to promote anabolic processes[44]. We also observed that GS exhibit increased intracellular levels of upstream glycolytic intermediates by metabolomics, though the rate of glycolysis (ECAR) was higher in DGC. These results suggest that glycolysis in GS may be stalled at the level of Enolase, and, moreover, that DGC are better positioned to circumvent the inhibition of mitochondrial respiration through enhanced

glycolysis. Genotype-dependent analysis of metabolism in DMG previously revealed elevated glycolysis in H3K27M gliomas compared to H3 wild-type tumors[24]. It will be important to test how the differentiation state interacts with the genotype to regulate glycolysis.

Along these lines, metabolites such as taurine, creatine, creatinine, uric acid, and hydroxyproline, which are reported to be associated with cellular differentiation of oligodendrocytes, cardiomyocytes, mesenchymal cells and adipocytes[45,65–67], were found to be upregulated in DGC. Indeed, taurine has been demonstrated to play a role in several biological processes, including the prevention of mitochondria damage, stabilization of OXPHOS in cardiomyocytes, and protection against endoplasmic reticulum (ER) stress. Creatine is involved in ATP buffering and enhancing mitochondria function[65]. These results suggest that these metabolites are pertinent to cellular differentiation processes, irrespective of the cell of origin.

Our transcriptomics analysis revealed AC-like DGC exhibited an enhanced mesenchymal phenotype. Based on this insight, we demonstrated that DGC were more sensitive to agents that promote ferroptosis. Conversely, OPC-like GS cells, whose gene signature correlated with higher disease aggressiveness and decreased overall survival in patients, upregulated cholesterol metabolism and mitochondrial OXHPOS. The upregulated sterol biosynthetic pathway in OPC-like GS cells could lead to enhanced production of squalene and/or 7-dehydrocholesterol, metabolites that promote resistance to ferroptosis, potentially explaining ferroptotic resistance of the GS state[68]. In either case, targeting these pathways with Phenformin and Metformin (mitochondria complex I inhibitors) or statins (sterol biosynthesis inhibitor) resulted in selective killing of GS compared to DGC in vitro. As proof of principle, we also demonstrated considerable in vivo activity of these metabolic inhibitors in an orthotopic mouse model of DMG.

In DIPG-007, SF7761, and murine PPK cell line pairs, GS populations could be selectively targeted by inhibiting OXPHOS. In contrast, DIPG-XIII cell line pairs showed a limited differential phenotype to OXPHOS inhibition. Therefore, future studies with these models could help to determine predictive biomarkers of sensitivity to OXPHOS targeting. It is conceivable that the limited differential phenotype between DIPG-XIII GS and DGC in outcomes such as cell proliferation, purine nucleotide pools, TCA cycle metabolites, OCR, SRC, energy charge, sensitivity to radiation, and sensitivity to OXPHOS inhibitors may result from oncogenic signaling related to *MYC* and *EGFR*. Indeed, the greater than two-fold upregulation of *MYC* and *EGFR* seen in DIPG-007 and SF7761 GS, in comparison to their DGC counterparts, was not similarly observed in DIPG-XIII GS vs. DGC. Along these lines, a question that merits future investigation is whether specific oncogenic signaling pathway(s) or transcription factor(s) operating in distinct tumor subpopulations direct metabolic reprogramming. For instance, MYC and EGFR have been reported to be critical for maintenance of the brain tumor-initiating cells in adult glioblastoma[69,70].

The concentration of cholesterol is highest in the brain, at approximately 20% of total body cholesterol[71]. In addition, the majority

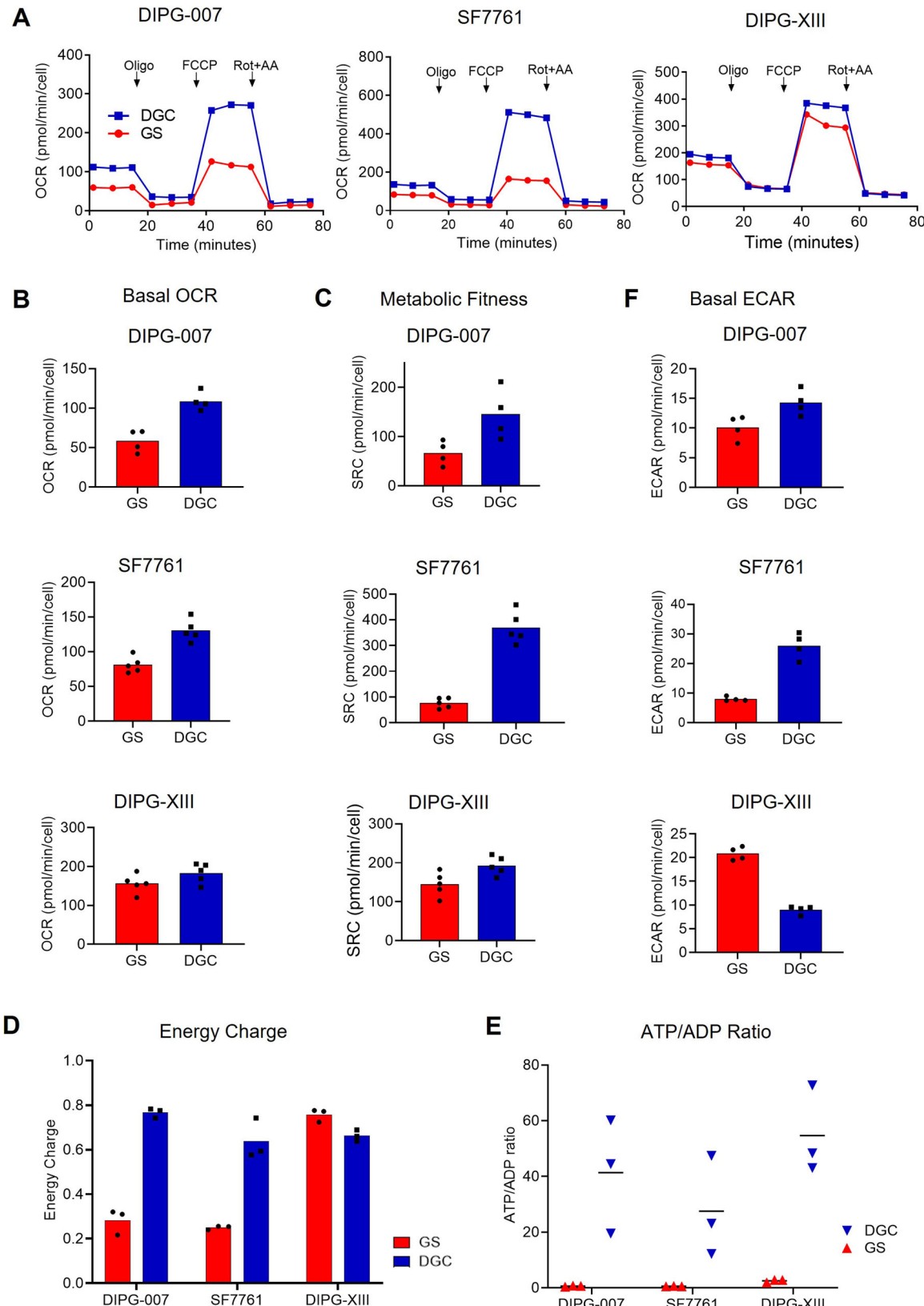

**Fig. 6 | Bioenergetic properties of oligodendrocyte precursor (OPC)-like gliomaspheres (GS) and astrocyte (AC)-like differentiated glioma cells (DGC).**
**A–C** Bioenergetics analysis (Seahorse assay, mitochondrial stress test) of DIPG-007, SF7761, and DIPG-XIII GS and DGC. (**A**) Oxygen consumption rate (OCR) during mitochondrial stress test; (**B**) Basal OCR, (**C**) Spare respiratory capacity (SRC). Determination of (**D**) energy charge calculated as [(ATP) + 0.5 (ADP)]/ [(ATP) + (ADP) + (AMP)], and (**E**) ATP/ADP ratios in GS and DGC across all three DIPG pairs. **F** Extracellular acidification rate (ECAR) of DIPG-007, SF7761, and DIPG-XIII GS and DGC. **A–C**, **F** $n = 2$ independent experiments. **D** Experiments were performed in technical triplicates and bar graphs are calculated values of energy charge expressed as mean; $n = 1$. **E** Experiments were performed in triplicates on the same day and expressed as mean of ATP/ADP ratio in GS vs. DGC, $n = 1$. Data in D and E were extracted from the metabolomics analysis presented in Fig. 2A.

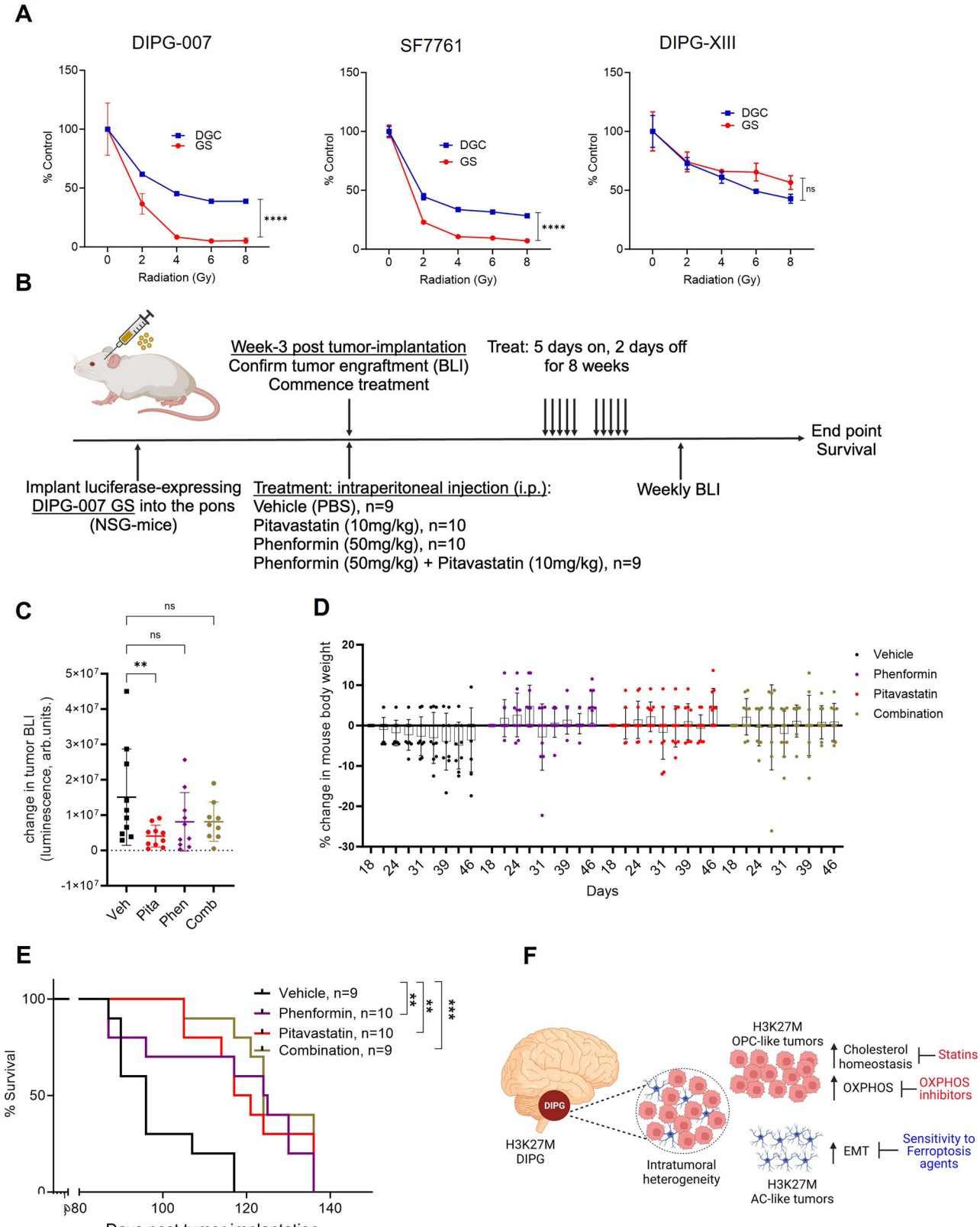

of brain cholesterol results from de novo synthesis, rather than uptake from circulation or peripheral tissues[72]. These results may provide mechanistic insight into the dependence of stem-like GS on cholesterol, and not on other outputs of the sterol biosynthesis pathway. Further, astrocytes are known to be the predominant producers and suppliers of cholesterol to other cells in the brain, including cancer cells[73]. Indeed, a dependency on cholesterol and the liver X receptors (LXR) axis as well as lanosterol synthase has been reported in brain tumors[74,75]. Our study, therefore, adds to the growing evidence of a metabolic dependency of brain tumors on cholesterol and specifically presents cholesterol targeting as a therapeutic inroad for stem-like and tumorigenic H3K27M midline gliomas.

**Fig. 7 | Statin and Complex I inhibitors prolong survival in an orthotopic model of DIPG. A** Dose-response curves of radiation-treated DIPG-007, SF7761, and DIPG-XIII gliomaspheres (GS) and differentiated glioma cells (DGC). Cell viability assessed at 7 days post-treatment using Cell Titer-Glo 2.0 (DGC) or 3D (GS) with results expressed as percent of control and error bars representing mean ± SD from 3 biological replicates; ns = not significant; *$p < 0.05$; **$p < 0.01$; ***$p < 0.001$; ****$p < 0.0001$**** as determined by area under proliferation curve test followed by two-tailed Student's t-test. **B** Schematic of in vivo experiment using a luciferase-expressing DIPG-007 orthotopic xenograft model and intraperitoneal administration of Pitavastatin (Pita) (10 mg/kg; $n = 10$ mice), Phenformin (Phen) (50 mg/kg; $n = 10$ mice), or the combination (Comb) ($n = 9$ mice) of both drugs at indicated doses or vehicle control (Veh) (PBS; $n = 10$). **C** Quantitation of change in tumor volume based on bioluminescence imaging (BLI). To calculate BLI, animals were injected with luciferin and luminescence was captured and quantitated. Data are presented as change in BLI between weeks 3 and 7 post-tumor implantation in arbitrary units (arb.units) with error bars representing mean ± SD and multiple comparisons made using uncorrected Fisher's LSD (Pita vs. control, **$p = 0.0073$). **D** Percent change in body weight of DIPG-007 tumor-bearing NSG mice in "C" following treatment, with error bars representing mean ± SD. **E** End-point survival analyzes of treatment and control tumor-bearing mice using Log-rank (Mantel-Cox) test (Pita vs. control, **$p = 0.0016$; Phen vs. control, **$p = 0.0097$; Pita + Phen vs. control, ***$p < 0.0001$). Vehicle, $n = 9$; Phenformin, $n = 10$; Pitavastatin, $n = 10$; Combination, $n = 9$. **F** Model of H3K27M DIPG intratumoral heterogeneity, indicating specific vulnerabilities within oligodendrocyte precursor (OPC)-like and differentiated astrocyte (AC)-like tumor populations and their respective targeting strategies. Gy, Gray; OXPHOS, oxidative phosphorylation. For panels A and F, red indicates GS; blue, DGC. For panels C–E, black indicates vehicle control; red, Pita; purple, Phen; olive, combination Pita + Phen. Figure 7B, F were created with BioRender.com released under a Creative Commons Attribution-NonCommercial-NoDerivs 4.0 International license.

Targeting DMG via OXPHOS and cholesterol inhibition is a promising strategy in that the inhibitors of these pathways are clinically approved drugs and have been evaluated as chemo-sensitization agents in cancer clinical trials[58,59]. Moreover, a number of statins are known to penetrate the blood brain barrier[63,64]. In addition, Metformin, a biguanide and analog of phenformin, has been used in the clinic for several decades, and it is currently being tested in several cancer clinical trials as a chemo-adjuvant. Importantly, it too displays some degree of brain penetrance[62]. A mechanistic caveat with biguanides is discerning the contribution from their dual therapeutic actions. As our in vitro studies demonstrate, biguanides are inhibitors of mitochondrial respiration; however, in organisms, biguanides also lower circulating glucose and insulin[76,77]. We propose that the anti-tumor activity observed in our in vivo models reflect direct cancer cell targeting, based on the concentrations of phenformin achieved in vivo relative to its efficacy in vitro. However, future studies will be required to determine the impact of blood glucose and insulin lowering in this context. In either case, we put forth that targeting mitochondria OXPHOS and cholesterol biosynthesis could potentially have immediate clinical utility for DMG patients. Lastly, given the limited combinatorial activity of OXPHOS inhibitors and statins, future studies will be required to test efficacy alongside ionizing radiation therapy.

Metabolic dependencies have been investigated in H3K27M gliomas in comparison to H3WT tumors or normal brain tissue, and these studies have revealed dependencies on glucose, glutamine and mitochondrial metabolism[24–26]. Our study investigates the differentiation-state dependent metabolic vulnerabilities in H3K27M DMGs and therefore adds to the growing body of work on DMG metabolism. Notably, our work presents actionable metabolic vulnerabilities that can be leveraged to develop treatment options for this devastating disease (Fig. 7F). With the recent clinical promise of ONC201, it is possible that combination therapies can be devised to target heterogenous populations of H3K27M DMG tumor cells. Indeed, the findings from this study are significant in that they provide a framework for future investigations that could, by extension, have broad implications in the rational design of precision treatment approaches for H3K27M DMG patients based on tumor composition and abundance of specific tumor cell-types.

## Methods
### Ethics statement
All animal procedures used in this study were approved by the University of Michigan Animal Care and Use Committee. Tumor size/burden was monitored every 2–3 days. In accordance with protocol, if animals became moribund or exhibited neurological symptoms, they were immediately euthanized.

### Cell lines and culture conditions
HSJD-DIPG-007 (referred to as DIPG-007, H3.3K27M) was obtained from Dr. Rintaro Hashizume, Northwestern University; RRID: CVCL_VU70. SU-DIPG-XIII (referred to as DIPG-XIII, H3.3K27M) was obtained from Dr. Michelle Monje, Stanford University; RRID: CVCL_6948. SF7761 (H3.3K27M) was purchased from Millipore Sigma (#SCC126). All cells were cultured in a humidified incubator at 37 °C and 5% $CO_2$. DIPG-007 and DIPG-XIII GS were cultured in base media containing equal parts Neurobasal-A Medium (Gibco; #10888022) and DMEM/F12 (Gibco; #11330032) with added HEPES (10 mM) (Gibco; #15630080), Sodium Pyruvate (1 mM) (Gibco; #11360070), MEM NonEssential Amino Acids Solution (1X) (Gibco; #11140050), GlutaMAX-I Supplement (1X) (Gibco; #35050061), and supplemented with fresh B-27 Supplement minus vitamin A (1X) (Gibco; #12587010), Heparin Solution (2 µg/mL) (StemCell Technologies; #07980), human-EGF (20 ng/mL) (Peprotech; #AF-100-15), human-bFGF (20 ng/mL) (Peprotech; AF-100-18B), PDGF-AA (10 ng/mL) (Peprotech; 100-13 A), and PDGF-BB (10 ng/mL) (Peprotech; #100-14B). SF7761 GS were cultured in a base media containing Neurobasal-A Medium with added N-2 Supplement (1X) (Gibco; #17502048), B-27 Supplement (1X) (Gibco; #17504044), L-glutamine (2 mM) (Gibco; #25030081) and supplemented with fresh Heparin Solution (2 µg/mL), human-EGF (20 ng/mL), human-bFGF (20 ng/mL), and BSA (45 ng/ml) (Sigma; #A8412).

PPK cells were generated as an In Utero Electroporation (IUE) murine model of H3K27M glioma. IUE was performed using sterile technique on isoflurane/oxygen-anesthetized pregnant C57BL/6 or CD1 females at E13.5. Tumors were generated with lateral ventricle (forebrain) introduction of plasmids: (1) PB-CAG-DNp53-Ires-Luciferase (dominant negative TP53), (2) PB-CAG-PdgfraD824V-Ires-eGFP (PDGFRA D842V), and (3) PB-CAG-H3.3 K27M-Ires-eGFP (H3K27M), and are therefore referred to as the "PPK" model[50]. PPK GS were cultured in base media containing Neurobasal-A Medium (1X), Sodium Pyruvate (1 mM), MEM Non-Essential Amino Acids (1X), L-glutamine (2 mM), Antibiotic-Antimycotic (1X) (Gibco; #15240096) and supplemented with fresh B-27 Supplement (1X), N-2 Supplement (1X), Heparin (2 µg/mL), EGF (20 ng/mL), and FGF (20 ng/mL).

Differentiated human and murine H3K27M glioma cells were generated by dissociating the respective gliomaspheres into single cells with Accutase (Innovative Cell Technologies; #AT104) and subsequently cultured and maintained in the respective base media supplemented with 10% FBS (Corning; #35-010-CV) for 14 days to generate a monolayer adherent culture.

All cell lines used were routinely tested for mycoplasma using MycoAlert PLUS (Lonza; #LT07-710) and were validated by STR profiling. For STR profiling, DGC and GS of DIPG-007, DIPG-XIII and SF7761 (2 million cells each) were collected, washed once with PBS, and snap

frozen for shipment to the Arizona Genetics Core for cell line authentication (https://azgc.arizona.edu/faq/cell-line-authentication). Briefly, genomic DNA was isolated using the Qiagen DNA Easy Blood and Tissue Kit (#69506) per the manufacturer's recommended protocol and was genotyped for 15 Autosomal STR loci and Amelogenin (X/Y) using the Promega PowerPlex 16 HS PCR kit (#DC2101). PCR products were separated by capillary electrophoresis using an AB 3730 DNA Analyzer. Electropherograms were analyzed from the.fsa files and allelic values assigned using Soft Genetics, Gene Marker Software Version 3.0.1. Cell authentication was verified using the reference databases ATCC, DSMZ, and JCRB, with a minimum 80% match threshold indicating a shared genetic history.

### Proliferation assay
DGC or GS in culture were dissociated into single cells with 0.25% Trypsin/EDTA (Gibco; #25200114) or Accutase, respectively. Following this, 200,000 cells were plated in 60 mm dishes, and cultured in their respective growth media for up to 12 days with media replenished every 3 days. At the indicated times, cells were again dissociated and counted using the Countess II FL Automated Cell Counter (Invitrogen) to assess cell proliferation.

### Drug treatment and viability assay
The following compounds used in this study were purchased from Cayman Chemicals: (1S,3 R)- RSL3 (RSL3, #19288), Ferrostatin-1 (#17729), z-vad-FMK (#14463), Necrosulfonamide (#20844), Bafilomycin A-1 (#11038), Metformin (#13118), Phenformin (#14997), IACS-010759 (#25867), Atorvastatin (#10493), Fluvastatin (#10010334), Pitavastatin (#15414), Mevalonate (#20348), Farnesyl Pyrophosphate (#63250), Geranylgeranyl Pyrophosphate (#63330), and Coenzyme Q10 (#11506). Cholesterol-Water Soluble (Cholesterol–methyl-β-cyclodextrin, #C4951) and Trolox (#238813) were purchased from Millipore Sigma. Equal numbers of DIPG-007, SF7761 and DIPG-XIII GS and DGC were plated in white opaque 96-well plates at 2000–3000 cells per well and incubated overnight. Cells were treated with compounds at the indicated concentrations and lengths of time described in the figure legends. At end point, an equal volume of Cell Titer-Glo 2.0 (Promega; #G9242) or Cell Titer-Glo 3D (Promega; #G9683) reagent was added to each well and viability assessed according to the manufacturer's protocol. Luminescence was detected and measured using a SpectraMax M3 (Molecular Devices, San Jose, CA) and data was analyzed with GraphPad Prism software.

### Cell culture irradiation
DGC cultures were seeded at 2000 cells per well in 96-well white-walled culture plates and allowed to adhere for 24hrs. GS cultures (2000 cells/well) were then added to these culture plates, which were then treated with increasing doses of radiation (0, 2, 4, 6, 8 Gy). Fresh media was spiked in all wells on day 4 of the experiment. For combination studies, DIPG-007 GS cultures (2000 cells per well) were seeded in 96-well white-walled culture plates and treated with increasing doses of Phenformin (0, 3, 10, 30, and 60 μM) at 24 h. Cells were then irradiated with 2 Gy either 24hrs or 72hrs later. Fresh Phenformin-treated media was spiked on day 4 immediately following radiation treatment. Cell viability was determined using Promega's CellTiter-Glo 3D Cell Viabilty Assay reagent at day 7 post-treatment using a SpectraMax M3 (Molecular Devices, San Jose, CA). Irradiations (2 Gy per min) were performed by the University of Michigan Experimental Irradiation Core using a Philips RT250 (Kimtron Medical), in accordance with literature precedent[78].

### Detection of lipid ROS
To assess levels of lipid ROS in cells, 200,000 DGC were plated in a 6-well plate overnight and treated with the indicated compounds. At end point, cells were washed twice with PBS and stained for 20 min

with 2 μM C11-BODIPY (Invitrogen; #D3861) in a phenol red-free media. Following staining, cells were washed twice with PBS and dissociated to single cells with trypsin. The cells were then transferred to round-bottom 96-well plates on ice, co-stained with Sytox-blue (Invitrogen; #S34857) to distinguish viable cells, and analyzed on a ZE5 Cell analyzer (Bio-Rad Laboratories, Hercules, CA). C11-BODIPY signal was captured with the FITC channel. Analysis of data was performed using FlowJo v.10 software.

### Seahorse bioenergetics assay
The cellular bioenergetic state was analyzed using a Seahorse XF-96 Extracellular Flux Analyzer (Agilent Technologies, Santa Clara, CA). The sensor cartridges were incubated in deionized water overnight, and on the day of the assay, the cartridges were hydrated in XF Calibrant Solution (Agilent; #103059-000) for 1 h in a non-$CO_2$ incubator at 37 °C. The hydrated cartridges were loaded with oligomycin (1 μM), FCCP (1 μM), rotenone (0.1 μM), and antimycin A (1 μM) to perform the Seahorse XF Cell Mito Stress Test (Agilent; #103015-100). Concurrently, 96-well Seahorse cell culture plates were coated overnight with laminin, and on the day of the assay, dissociated DIPG GS were washed and resuspended in Seahorse XF DMEM medium (Agilent; #103575) supplemented with XF Glutamine Solution (2.5 mM) (Agilent; #103579), XF Glucose Solution (17.5 mM) (Agilent; #103577), XF Sodium Pyruvate Solution (1 mM) (Agilent; #103578), MEM Non-Essential Amino Acids Solution (1X) (Gibco), human-EGF (20 ng/mL) (Peprotech), human-bFGF (20 ng/mL) (Peprotech), and Heparin (2 μg/mL) (StemCell Technologies). Dissociated DGC were washed and resuspended in Seahorse XF DMEM medium (Agilent) supplemented with similar concentrations of XF Glutamine Solution, XF Glucose Solution, SXF Sodium Pyruvate Solution, and MEM Non-Essential Amino Acids Solution. DGC and GS single cells (150,000 to 200,000 cells) were seeded on laminin-coated plates and allowed to equilibrate for 30 min in a non-$CO_2$ incubator at 37 °C. Data were then acquired on the Seahorse analyzer. Following data acquisition, measurements were normalized based on cell number using the CyQuant NF Cell Proliferation Assay (Invitrogen; #C35006). For the Mito stress test, the basal oxygen consumption rate (basal OCR) was determined based on basal OCR measurements taken prior to addition of inhibitors. The spare respiratory capacity (SRC) was determined by subtracting basal OCR from maximal OCR measurements. Seahorse analysis was performed using the Wave 2.3 software.

### Metabolomics
**Metabolite extraction.** To generate intracellular metabolite fractions, an equal number of GS and DGC were cultured in 6-well plates for 36 h. Growth media was then removed, cells were lysed with ice-cold 80% methanol on dry ice for 20 min, and lysates were collected and clarified by centrifugation. The metabolite load of intracellular fractions was normalized to protein content of parallel samples, and these volumes were then lyophilized in a Savant SPD1030 SpeedVac (Thermo-Scientific). Dried metabolite pellets were resuspended in 50:50 mixture of HPLC-grade methanol:d$H_2O$ and subjected to metabolomics analysis.

**LC/MS-based Snapshot Metabolomics.** LC/MS-based Metabolomics was performed on an Agilent 1290 Infinity II LC-coupled to a 6470 Triple Quadrupole (QqQ) tandem mass spectrometer (MS/MS)[79,80]. Briefly, Agilent Masshunter Workstation Software LC/MS Data Acquisition for 6400 Series Triple Quadrupole MS with Version B.08.02 was used for compound optimization, calibration, and data acquisition. The QqQ data were pre-processed with Agilent MassHunter Workstation QqQ Quantitative Analysis Software (B0700). Two-tailed t-test with a significance threshold level of 0.05 was applied to determine statistical significance between conditions. Graphs were generated using GraphPad Prism software. Heatmaps were generated and data

clustered using Morpheus Matrix Visualization and analysis tool (https://software.broadinstitute.org/morpheus). Pathway analyses were conducted using MetaboAnalyst (https://www.metaboanalyst.ca). Unprocessed Snapshot metabolomics data are provided in Source Data File 2.

**U13C-Glucose isotope tracing.** As a glucose-free version of DMEM/F12 was not commercially available, glucose-free DMEM (Gibco; #A10443001) and glucose-free Neurobasal-A Medium (Gibco; #A2477501) replenished with equivalent concentrations of glutamine and sodium pyruvate were used to prepare the 1:1 glucose-free isotope tracing base media used in this assay. DIPG-007 GS and DGC were seeded at 300,000 cells per 6-well for 24hrs in their respective growth media. DIPG-007 GS-containing media was then collected and GS cells were spun down and resuspended in 2 mL supplemented isotope tracing media containing 21.25 mM of either uniformly labeled 13C-glucose (Cambridge Isotope Laboratories; #CLM-1396-5) or 12C-glucose (Sigma; #G7528). The same amount of isotope tracing base media supplemented with 10% dialyzed FBS (Cytiva; #SH30079.03) containing either 21.25 mM uniformly labeled 13C-glucose (Cambridge Isotope Laboratories) or 12C-glucose (Sigma) was added to DIPG-007 DGC cells following aspiration of the plating media. Metabolite extraction was performed at 0.25, 1, 4, and 24 hour time points using 80% cold MeOH. Samples were normalized to the protein content of a parallel sample and were then lyophilized via Speed Vac. The isotope tracing experiments utilized the same chromatography as described in the Snapshot Metabolomics section, which was coupled to an Agilent Q-TOF 6545 mass spectrometer. LC/MS was performed as previously described[80]. Data processing was performed in Agilent MassHunter Workstation Profinder 10.0 Build 10.0.10062.0. Isotopologue distributions were derived from a compound standard library built in Agilent MassHunter PCDL (Personal Compound and Database Library) v7.0.

## Lipidomics
DIPG-007 DGC and GS cultures with U13C-glucose tracing were prepared as described above. Lipid extraction was performed as described by Bielawski[81] with slight modifications. Samples were thawed on ice and then samples were extracted with 1.0 mL of IPA:Water:EtOAc (30:10:60, v:v:v) and internal standard mixture of EquiSPLASH™ LIPIDOMIX® Quantitative Mass Spec Internal Standard (Avanti Polar Lipids, Birmingham, AL). The extract was vortexed and sonicated for 2 min, followed by centrifugation for 10 min at 8000× $g$ at 4 °C. The organic upper phase was transferred to a new tube. The pellet was re-extracted with an additional 0.5 mL of IPA:Water:EtOAc (30:10:60, v:v:v). The supernatants were combined and placed at −20 °C for 24 h. The supernatants were dried down using a speed vac. The dried sample was reconstituted in 150 μL Solvent A. The suspension was vortexed for 5 minutes and then centrifuged for 10 min at 17,000 × $g$ at 4 °C. The supernatant was transferred to an auto-sampler vial for UHPLC-MS analysis.

Lipid profiling was conducted using a Vanquish UHPLC system with an Orbitrap Fusion Lumos Tribrid™ mass spectrometer using a H-ESI™ ion source (all Thermo Fisher Scientific, Waltham, MA) with a Waters ACQUITY UPLC CSH C18 column (150 × 1 mm, 1.7 μm particle size, Milford, MA). Solvent A was HPLC grade water:acetonitrile (40:60, v:v) with 0.1% formic acid and 10 mM ammonium formate. Solvent B was HPLC grade isopropanol:acetonitrile (95:5, v:v) with 0.1% formic acid and 10 mM ammonium formate. The column was maintained at 65 °C and a flow rate was set at of 110 μL /min. The gradient of the solvent B is 15% (B) at 0 min, 30% (B) at 2 minutes, 2−2.5 min 48% (B), 2.5−11 min 82% (B), 11−11.01 min 99% (B), 11.01−12.95 min 99% (B), 12.95−13 min 15% (B), and 13−15 min 15% (B). Data acquisition was carried out in positive charge mode, with the ion source spray voltage configured to 4000 V. The mass spectrometry analysis spanned a scan range of 200−1000 m/z for the full scan, and the MS1 resolution was

established at 500 K at m/z 200. The AcquireX mode was employed for the MS2 acquisition, which was performed with a stepped collision energy of 30%, along with a 5% spread for the fragment ion MS/MS scan.

Raw data was converted into mzML format using Proteowizard mscovert software[82]. MS-DIAL software (verion 4.9.221218)[83] was used for general lipidomics data analysis including compound identification with LipidBlast, which is default library in MS-DIAL.

## Energy charge calculation
For each sample, the ion current from the LC/MS analysis for adenosine triphosphate (ATP), adenosine diphosphate (ADP), and adenosine monophosphate (AMP) levels were enumerated, and the adenylate energy charge (AEC) was calculated by applying the formula [(ATP) + 0.5(ADP)]/ [(ATP) + (ADP) + (AMP)]. The ratio of ATP/ADP was evaluated by directly determining the ratio of ATP to ADP metabolite levels in each cell line.

## RNA sequencing
Total RNA was extracted from DIPG-007, SF7761, and DIPG-XIII GS and DGC using the RNeasy Mini Kit (Qiagen) according to the manufacturer's instructions. Strand-specific, poly-A+ libraries were prepared using NEBNext Ultra II Directional RNA Library Prep Kit (New England Biolab; #E7760L), the Poly(A) mRNA Magnetic Isolation Module (New England Biolab; #E7490L), and NEBNext Multiplex Oligos for Illumina Unique Dual (New England Biolab; #E6440L). Sequencing was performed on the NovaSeq-6000 (Illumina), yielding 150-base, paired-end reads. Library preparation and sequencing were performed by the University of Michigan Advanced Genomics Core (Ann Arbor, MI).

The reads were trimmed using Trimmomatic v0.36[84] and the library qualities were assessed using FastqQC v0.11 for trimmed reads (https://www.bioinformatics.babraham.ac.uk/projects/fastqc/). RSEM v1.3.1 and STAR v2.5.2a were used to generate paired-end alignments and counts[85,86].

Differential gene expression analysis was performed using DESeq2 v1.26.0 and the *apeglm* shrinkage estimator was used to adjust log$_2$ fold-changes[87]. Normalized counts were obtained DESeq2 (default method; median of ratios). Differentially expressed genes were defined as having adjusted $p$-value < 0.05 and fold change > 1.5 (up or down). Variance stabilized transform (VST) gene counts were used in principal component analysis to identify the major sources of variance and evaluate the similarity of replicates. The reference sequence hg38 (GRCh38) and annotations, including gene IDs, were obtained from GENCODE v29. Differentially expressed genes were analyzed using GSEA using the HALLMARK gene sets.

## DIPG and DMGs dataset analysis
The human DIPG and DMG dataset was mined from Mackay et al.[40]. The downloaded patient samples contained $n = 78$; Female=41, Male=28, unknown=9. Age range was 1.2−17.9 years (average 9.8 ± 12 years). Expression levels of GS-related genes in 76 H3K27M diffuse midline gliomas were segregated into high vs. low gene expression categories using unbiased K-means clustering ($K = 2$ to assign two groups). Kaplan-Meier analysis was then performed between high (defined as upper quartile) vs. low (all remaining samples) tumors to determine differences in overall survival. Data were analyzed by the Log rank test.

## Western blot
To assess protein levels, DIPG-007, SF7761, DIPG-XIII GS and DGC were lysed in RIPA buffer (Sigma; #R0278) containing protease (Roche; #04693132001) and phosphatase (Sigma; #P5726) inhibitors. Protein concentrations from whole cell lysates were determined using the Pierce BCA Protein Assay Kit (#23227), according to the manufacturer's protocol. Equal amounts of protein were subjected to separation on SDS-PAGE and transferred to a methanol-activated PVDF membrane.

Membranes were blocked with 5% milk in TBST (Tris-buffered saline containing 0.1% Tween 20) followed by incubation with primary antibodies diluted in 5% milk or BSA TBST at 4 °C overnight. The following primary antibodies and dilutions were used: OLIG2 (Cell Signaling Technology (CST); #65915; 1:1000), PARP (CST; #9542; 1:1000), HSP90 (CST; #4874; 1:10,000) and alpha-TUBULIN (clone 11H10, CST; #2125; 1:10,000). Following primary antibody incubation, the membranes were washed 3 times with TBST and incubated with species-appropriate secondary antibodies conjugated to horseradish peroxidase (HRP) at 1:10,000 dilution for 1 h at room temperature. Membranes were then washed 3× with TBST and chemiluminescence was detected using Clarity (Bio-Rad; #1705060) or Clarity Max (Bio-Rad; #1705062) ECL substrate. The signal was captured with a Bio-Rad ChemiDoc imager and analyzed using Image Lab software.

## Mouse studies

Animal experiments were performed after approval from the University of Michigan Committee on Use and Care of Animals and were conducted as per NIH guidelines for animal welfare. All animal procedures were approved by Institutional Animal Care & Use Committee (IACUC) at the University of Michigan (IACUC approval # PRO00008865). Animals were housed and cared for according to standard guidelines with free access to standard diet (irradiated 5Lod (LabDiet)) and water ad libitum at constant ambient temperature and a 12-hour light cycle. All experiments were performed on NOD-SCID-IL2R gamma chain-deficient (NSG) mice that were 8–10 weeks old, with males and females used equally. Mice, including littermates of the same sex, were randomly assigned to control or treatment conditions. All animal experiments were performed in a blinded manner.

## In vivo xenograft tumor studies

Luciferase-expressing DIPG-007 GS (400,000 cells) suspended in 2 µl PBS were injected into the pons to establish orthotopic xenografts under anesthesia, as follows. NSG mice were anesthetized with 75 mg/kg dexmedetomidine and 0.25 mg/kg ketamine by intraperitoneal injection. Carprofen (5 mg/kg) was used for analgesia. Mice were mounted on a stereotaxic device. A small sagittal incision was made using a scalpel and a small hole was created using a micro drill at 1.0 mm posterior and 0.8 mm lateral left from lambda. A sterile Hamilton syringe was used to inject cells. Half of the cells were injected at 5 mm depth from the inner base of the skull and the remaining cells were injected after 0.5 mm retraction in order to implant cells into the pontine tegmentum. After surgery, 1 mg/kg atipamezole solution was intraperitoneally injected for anesthesia reversal. Tumor engraftment was confirmed by bioluminescence imaging. Treatment and controls groups were delineated in a random fashion after mice developed tumors. Tumor monitoring with bioluminescence was performed in a blinded manner. Treatment commenced 3 weeks post-tumor implantation. The mice were randomized into 4 groups receiving either vehicle (PBS), Pitavastatin (10 mg/kg), Phenformin (50 mg/kg), or combination Pitavastatin (10 mg/kg) and Phenformin (50 mg/kg). The drugs were administered intraperitoneally using a 5-day on/2-day off course for 9 weeks. Tumor size was measured using bioluminescent imaging (IVIS) up to 10 weeks post-implantation, at which mice were then monitored for end-point survival. At end point, mice were euthanized via carbon dioxide and brain tissues harvested for immunohistochemistry analysis.

## Bioluminescent imaging

$4 \times 10^5$ DGC or GS DIPG-007 cells were suspended in 2 µL of media and injected into the cerebella of NSG mice, as described above. Tumor engraftment was confirmed by bioluminescence imaging. Luciferin (100 uL of a 15 mg/mL dilution in PBS) was injected intraperitoneally into each mouse. Mice were anesthetized and placed into the IVIS

Spectrum In Vivo Imaging System (PerkinElmer) and imaged, as described previously[24]. The maximum tumor volume permitted was 8 mm × 3 within the mouse pons.

## Immunohistochemistry

Tissue sections on slides were deparaffinized with Histo-Clear (National Diagnostics; #HS200) and re-hydrated with graded ethanol and water. Sections were quenched with 1.5% hydrogen peroxide in 100% methanol for 15 min at room temperature. Antigen retrieval was performed in sodium citrate buffer (2.94 g sodium citrate, 500 µl Tween 20, pH 6.0) and slides were maintained at a rolling boil for 20 min. After cooling, tissue sections were blocked in 2.5% bovine serum albumin (Sigma; #A2153), 0.2% Triton X-100, in PBS for 1 h at room temperature. Sections were incubated overnight at 4 °C with primary antibodies diluted in blocking buffer: anti-OLIG2 (CST; #65915; 1:100), anti-KI67 (Abcam; #ab15580; 1:1000), anti-CC3 (CST; #9664; 1:100), anti-GFAP (Dako; #Z0334; 1:1000), anti-S100 (Dako; #GA50461-2; 1:200). After rinsing with PBS, slides were incubated for 1 h at room temperature with secondary antibodies diluted in blocking buffer. Endogenous biotin, biotin receptors, and avidin binding site blocking was performed using the Avidin/Biotin Blocking Kit (Vector Laboratories; #SP-2001) per manufacturer's protocol. Substrate reaction and detection was performed using DAB Peroxidase (HRP) Substrate Kit (With Nickel), 3,3'-Diaminobenzidine (Vector Laboratories; #SK-4100) as detailed per the manufacturer's protocol. Slides were counterstained with Mayer's hematoxylin solution, mounted in Permount Mounting Medium (Fisher Chemical; #SP15-100), coverslipped, and allowed to dry overnight before imaging.

Stained slides were imaged with CellSens Standard software using an Olympus BX53F microscope, fitted with an Olympus DP80 digital camera (Olympus Life Science). 3–5 images were taken per slide at 20x magnification and positive DAB signal was quantified using QuPath software. Graphs were generated and statistics were calculated using GraphPad Prism. Data in graphs are presented as the mean with standard deviation, including individual datapoints that represent an average for each mouse. Data were analyzed with a one-way ANOVA, Tukey's multiple comparisons test.

## Statistical Analysis

Statistical analyzes were performed using GraphPad Prism (Graph Pad Software Inc). For assessment of statistical significance between treatments and stimuli in growth curves, area under the curve (AUC) was employed. Two-group comparisons were analyzed using the unpaired two-tailed Student's t-test. Error bars represent mean ± standard deviation, unless noted otherwise, and the significance annotations are indicated in figure legends and Source data file. For all figures, $p$-value < 0.05 was considered statistically significant. The number and type of experimental replicates as well as the explanation of significant values are indicated in the figure legends.

## Reporting summary

Further information on research design is available in the Nature Portfolio Reporting Summary linked to this article.

# Data availability

RNAseq data are deposited at the GEO public repository using the following identifier, GSE197145. Raw metabolomics data are provided as Supplementary Data 2. The raw tracing data are provided in Supplementary Data 3. Patient data were reanalyzed from Mackay et al.[40]. This study pooled data from EGA, Gene Expression Omnibus, and ArrayExpress; links below. Source data file is provided with this manuscript. European Genome-Phenome Archive (EGA): EGAS00001000226, Mackay, et al.[40] https://ega-archive.org/datasets/EGAD00001000134. EGAS0000100192, Parker, et al.[88] https://ega-archive.org/studies/EGAS00001000192. EGAS00001000575,

Buczkowicz, et al.[89] https://ega-archive.org/studies/EGAS00001000575. EGAS00001000720, Fontebasso, et al.[90], https://ega-archive.org/datasets/EGAD00001000792. EGAS00001001139, Mackay, et al.[40] https://ega-archive.org/datasets/EGAD00001002006. Gene Expression Omnibus (GEO): GSE19578, Paugh, et al.[91] https://www.ncbi.nlm.nih.gov/geo/query/acc.cgi?acc=GSE19578. GSE26576, Paugh, et al.[92], https://www.ncbi.nlm.nih.gov/geo/query/acc.cgi?acc=GSE26576. GSE21420, Barrow, et al.[93], https://www.ncbi.nlm.nih.gov/geo/query/acc.cgi?acc=GSE21420. GSE34824, Schwartentruber, et al.[6], https://www.ncbi.nlm.nih.gov/geo/query/acc.cgi?acc=GSE34824. GSE36245, Sturm, et al.[94], https://www.ncbi.nlm.nih.gov/geo/query/acc.cgi?acc=GSE36245. GSE36278, Sturm, et al.[94], https://www.ncbi.nlm.nih.gov/geo/query/acc.cgi?acc=GSE36278. GSE50022, Buczkowicz, et al.[89], https://www.ncbi.nlm.nih.gov/geo/query/acc.cgi?acc=GSE50022. GSE50021, Buczkowicz, et al.[89], https://www.ncbi.nlm.nih.gov/geo/query/acc.cgi?acc=GSE50021. GSE50024, Buczkowicz, et al.[89], https://www.ncbi.nlm.nih.gov/geo/query/acc.cgi?acc=GSE50024. GSE55712, Fontebasso, et al.[90], https://www.ncbi.nlm.nih.gov/geo/query/acc.cgi?acc=GSE55712. ArrayExpress: E-TABM-857, Payne, et al.[95], https://www.ebi.ac.uk/biostudies/arrayexpress/studies/E-TABM-167. E-TABM-1107, Puget, et al.[96], https://www.ebi.ac.uk/biostudies/arrayexpress/studies/E-TABM-1107 Source data are provided with this paper.

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

## Acknowledgements

This work was funded by a joint Defeat DIPG and ChadTough Foundation fellowship award (NEM); Alex's Lemonade Stand Foundation POST Grant (DM); Alex's Lemonade Stand Foundation (SV and CAL); University of Michigan Pediatric Brain Tumor Initiative (CK, SV, and CAL); R01NS110572, R01CA261926, 1R01NS127799 (SV); R01NS124607, R01NS110572 and DOD grant (CA201129P1) (CK). Metabolomics studies performed at the University of Michigan were supported by the Charles Woodson Research Fund and the UM Pediatric Brain Tumor Initiative. Research reported in this publication was also supported by the National Cancer Institutes of Health under Award Number P30CA046592 using the following Cancer Center Shared Resource(s): Flow Cytometry Core, Tissue and Molecular Pathology Core. Schematics and models were created using Biorender.com.

## Author contributions

NEM and CAL conceived of and designed this study. NEM, CJK, SV and CAL planned and guided the research. NEM, ALM, PS, CC, JKT, HSH, HG, DD, ZCN, MS, SRS, DDM, BC, LZ, BM, ZZ, MR, BL, VNY, IK performed experiments, analyzed, and interpreted data. NEM, ALM, ADP, DRW, LF, SA, CJK, SV, and CAL were involved in the conceptual design of experiments and proofreading of manuscripts. NEM and CAL wrote the manuscript. SV and CAL supervised the work carried out in this study.

## Competing interests

In the past three years, C.A.L. has consulted for Astellas Pharmaceuticals, Odyssey Therapeutics, Third Rock Ventures, and T-Knife Therapeutics, and is an inventor on patents pertaining to K-Ras regulated metabolic pathways, redox control pathways in pancreatic cancer, and targeting the GOT1-ME1 pathway as a therapeutic approach (US Patent No: 2015126580-A1, 05/07/2015; US Patent No: 20190136238, 05/09/2019; International Patent No: WO2013177426-A2, 04/23/2015). All other authors declare no competing interests.

## Additional information

[1]Chad Carr Pediatric Brain Tumor Center, University of Michigan, Ann Arbor, USA. [2]Department of Molecular & Integrative Physiology, University of Michigan, Ann Arbor, USA. [3]Graduate Program in Cancer Biology, University of Michigan, Ann Arbor, USA. [4]Laboratory of Brain Tumor Metabolism and Epigenetics,

Department of Pathology, University of Michigan Medical School, Ann Arbor, USA. [5]Department of New Biology, Daegu Gyeongbuk Institute of Science and Technology (DGIST), Daegu, South Korea. [6]Department of Surgery, University of Michigan, Ann Arbor, USA. [7]Graduate Program in Immunology, University of Michigan, Ann Arbor, USA. [8]Graduate Program in Molecular & Cellular Pathology, University of Michigan, Ann Arbor, USA. [9]Neuroscience Graduate Program, University of Michigan, Ann Arbor, USA. [10]Graduate Program in Cellular and Molecular Biology, University of Michigan, Ann Arbor, USA. [11]Department of Biostatistics, School of Public Health, University of Michigan, Ann Arbor, USA. [12]Graduate Program in Chemical Biology, University of Michigan, Ann Arbor, USA. [13]The Department of Pediatrics, Children's Mercy Research Institute (CMRI), Kansas, USA. [14]Huck Institutes of the Life Sciences, Pennsylvania State University, University Park, USA. [15]Department of Biochemistry and Molecular Biology and Department of Veterinary and Biomedical Sciences, the Pennsylvania State University, University Park, USA. [16]Department of Radiation Oncology, University of Michigan Medical School, Ann Arbor, USA. [17]Department of Pediatrics, University of Michigan Medical School, Ann Arbor, USA. [18]University of Pittsburgh Hillman Cancer Center, Pittsburgh, USA. [19]Department of Pathology, University of Michigan Medical School, Ann Arbor, USA. [20]Department of Internal Medicine, Division of Gastroenterology and Hepatology, University of Michigan, Ann Arbor, USA. [21]Rogel Cancer Center, University of Michigan, Ann Arbor, USA. ✉e-mail: svenneti@med.umich.edu; clyssiot@umich.edu

