## [Peer Review File · Nature Communications]

Therapeutic targeting of differentiation-state dependent metabolic vulnerabilities in diffuse midline gliomaREVIEWER COMMENTS

Reviewer #1 (Remarks to the Author): Expert in DIPG, RNA-seq, and proteomics

The authors of "Therapeutic targeting of differentiation state-dependent metabolic vulnerabilities in DIPG" employed a systems wide approach to characterize in vitro cellular heterogeneity of H3K27M+ using isogenic in vitro models comparing stem-like glioma cells (resembling oligodendrocyte precursors -OPC) and more differentiated astrocyte (AC)-like DIPG cells. In doing so they have identified metabolic programs operative in the different in vitro differentiation states, helping to identify treatment vulnerabilities. The potential for targeting metabolism may go some way to combat the inter- and intratumoral heterogeneity of the disease if similar influences are seen in vivo. The authors have revealed the potential of several clinically relevant metabolic drugs, important because as it stands, outcomes for patients diagnosed with DIPG remain are unacceptably poor. I have several major points that need to be addressed.

Major:

The authors describe cell populations as either isogenic GS or isogenic DGC. Can the authors describe whether they in fact know they are isogenic following the change in cellular state? Given studies confirming the intratumoral heterogeneity of patient tissues and cell lines, approximately 6 subclones are present in each tumor / cell line (PMID: 29967352; PMID: 26727948), so it is indeed possible that the change in cell culture condition helps to enrich for colonies/clones of particular genotype and cellotype?

In the PCA analysis of RNA expression data in Supp Fig. 1, the authors use PC2 vs. PC3 distinguish GS from DGC cells, suggesting that PC3 is more influenced by the cellular differentiation status than the cell type. For the most part this is true, except for SU-DIPG-13 which clusters independent of +/-serum. Remarkably, 1329 genes and 1163 genes were commonly differentially up- or down- regulated, GS vs. DGC respectively, across all three cell types. Given, serum fails to change the cellotype or principal components of SU-DIPG-13 it is interesting that so many genes are conserved compared to cells where serum induces such a pronounced change in growth and differentiation (PCA)? It is important to note that SU-DIPG-13 were established from an autopsy, of a frontal lobe metastasis, whereas the SU-DIPG-13P* were established from the pontine region of the same patient (PMID: 28434841). Given changes in EMT encourages metastatic dissemination, and the importance of the tumor microenvironment on transcriptional selection pressures that regulate growth, metabolism, differentiation, immune recognition, the recognition that these cells are different may go way to describe the sensitivity differences shown SU-DIPG-13 vs. SU-DIPG-13P*. Do SU-DIPG-13P* cells show a visual differentiation change +/- serum? If so, do GS established from these cells show the predictive markers of OPC/GS-like vs DGC/AC, revealed? Can the authors provide a summary of expression differences between the cell types that may help to describe why SU-DIPG-13 shows no change in morphology or growth rate? It may be possible that these metastatic cells are more amenable to other therapies shown to work in brain metastasizes, that fail in primary brain tumors.

I am wondering what would be the effect of the addition of serum if the cells were grown in ultra-low adherent flasks?

The authors provide interesting stem and differentiation markers GS/DGS and list individual gene expression profiles for each of cell lines in Supplementary Fig. S2. These data highlight the conserved expression of OLIG1 and OLIG2 in GS across cell types. However, the authors have neglected to provide any statistics here.

PDGFRA and BIM1 expression have previously been associated with oligodendrocyte precursor cells (OPC-like) from primary tumors. The data here shows GS expression profiles also enrich for the OPC-like gene expression signature identified by single cell RNA sequencing (PMC5949869). It is interesting that differentially, SOX2 and EGFR showed increased expression in cell types that show a growth and morphology change i.e. GS vs DGS, however not in SU-DIPG-13, which remained morphologically (transcriptionally?) the same regardless of GS vs DGS status. Given cell culture media is supplemented with EGF it would be interesting to see if the changes in morphology and growth rates are affected by the availability of hu-EGF (and potentially hu-PDGF-AA and hu-PDGF-BB, given no change in PDGFRA was seen in GS cells). It is well characterized that EGFR signaling promotes SOX2 expression, which in turn binds to the EGFR promoter to directly upregulates EGFR expression, providing a plausible link describing the differences in cellotype shown herein.

The clinical relevance of the GS gene expression profiles was investigated and correlated with OS data provided by a patient dataset. Given the differences in expression profiles in the SU-DIPG-13, I am interested to know if the authors were to exclude these data and mine GS high transcripts assessed just from DIPG007 and SF7761 GS cekk, what would the survival outcome look like? Each of the cell lines used are H3.3, with the Kaplan-Meier survival prediction looks remarkably

like H3.3K27M vs. H3.1K27M? Can the provide more statistics for Fig. 1H including median OS, CI Log Rank test results?

The glycolytic and glutamine metabolism signatures enriched in GS vs. DGC are also very interesting. As the authors highlight metabolic dependencies have been investigated in H3K27M gliomas revealing dependencies on glucose and glutamine metabolism. Did the RNA sequencing show changes in the expression of GLUT receptors or glutamine transporters in GS cells? Can modulation of glucose or glutamine change morphology?

Fig 3C shows GPX4 inhibition to cause cell death selectively in GDC cells at low concentration of RSL3. However, SF7761 GDC and GS show a similar IC50 between to RSL3. Do these cells show transcriptional differences that might help to explain the change in sensitivity seen? I am also interested to know if GDC cells show a more immunogenic transcriptome?

Does targeting cholesterol biosynthesis or OXPHOS change the differentiation status (GS) or morphology? Likewise, does enhancing ferroptosis drive dedifferentiation?

Ionizing radiation experiments in Figure 5G, require statistical evaluation? Given the mitochondrial spare respiratory capacity (SRC) result and the clinical relevance of metformin/phenformin, I am wondering about the combination index when used in combination with radiotherapy?

Further, cholesterol biosynthesis and mitochondrial SRC results are potentially very clinically relevant given the in vitro sensitivity of all cells to atorvastatin/pitavastatin, and DIPG 13P*, 007 and SF7761 to phenformin/metformin (metformin-SF7761?). CNS multiparametric optimization suggests that pitavastatin scores 4.81/6 and phenformin scores 3.57/6, highlighting the use of pitavastatin in vivo and potential of phenformin to cross the blood brain barrier, but not in high concentrations (PMID: 26991242). Do the authors hypothesize that modulation of glucose homeostasis is playing a role in the survival extension shown?

The authors indicate that (lines 404-405) "Notably, treatment with pitavastatin or phenformin resulted in the reduction in tumor volume based on BLI". Can the authors please provide more clarity to Figure 6C?

The single agent PDX results are very encouraging, however, please include more clear statistical evidence. Previous studies using glioma stem cells tested phenformin using both in vitro and in vivo models. Phenformin inhibited the self-renewal of glioma stem cells decreased the expression of stemness and mesenchymal markers, however at twice the concentrations (PMID: 27486821), these studies tested phenformin in PDX models using 100 mg/kg. Given the dose dependent decrease in proliferation of DIPG007 GS in vitro, and the increased cleaved PARP shown in Figure 4 (I acknowledge that at the same concentration no increase in in vitro apoptosis was shown in glioblastoma stem cells- PMID: 27486821), I am wondering why the authors chose to use this dose in their in vivo experiments?

Given that the majority of successful PDX models are established from neurosphere lines, DIPG007, DIPG13, SU-DIPG-13P*, it would be interesting to know whether DIPG007 cells induced to grow as GDC engraft?

Given the mice still succumb what is the differentiation status of the tumors at endpoint treatments vs. vehicle?

Additionally, given the data provided shows no significant change in animal weight, or BLI, with phenformin alone or in combination with pitavastatin, it would be important to know whether the treatments caused apoptosis in vivo, and if not at these concentrations, what dose can be achieved to cause DIPG specific apoptosis? Did the authors see any lactic acidosis in their animals?

Minor –

The DIPG acronym in the title should be expanded.

Word alignment in Figure 1A needs addressing

Nomenclature for proteins listed in captions needs addressing.

In the PCA analysis of RNA expression data in Supp Fig. 1, can the authors please inform why there are three (S1B) and four (S1C) individual results for SF7761 samples when there are only two comparisons in each PCA?

Statistics from Fig. 3A are missing.

Mouse weights should be presented as % change from baseline, rather than the actual weight.

Add unit of measurement in BLI experiment Figure 6C.

The survival analysis requires the inclusion of more statistical evidence.

The methods lack a description of SU-DIPG-13P*.

Reviewer #3 (Remarks to the Author): Expert in cancer metabolomics and metabolic vulnerabilities

In this manuscript by Mbah et al., the authors sought to investigate the metabolic programs that

operate in diffuse intrinsic pontine gliomas (DIPG), with the goal of identifying metabolic vulnerabilities for this deadly disease. The authors established in vitro isogenic models from three patient-derived H3K27M DIPG models in two distinct cellular states: 1) less-differentiated or stem-like gliomaspheres (GS) or 2) differentiated glioma cells (DGC). Using transcriptomics and metabolomics in the GS versus DGC cell states, the authors find that less-differentiated cells (GC) enrich for gene and metabolic signatures related to OXPHOS and cholesterol metabolism and that these GC cells were more vulnerable to inhibition of OXPHOS or statin treatment both in vitro and in an orthotopic model of DIPG. Interestingly, the more differentiated astrocyte (AC)-like cells showed a mesenchymal phenotype and these cells displayed sensitivity to ferroptosis.

This study suggests that metabolic inhibitors of OXPHOS or statins could be used to target DIPG for therapeutics. The study is interesting and clearly presented. However, I have some reservations about how generalizable these metabolic programs are for DIPG, given that the study is based on three cell lines. Moreover, the conclusions of the metabolic programs regulated in various cellular states rely primarily on steady-state metabolomics and transcriptomics data. The lack of metabolic tracing measurements, or proteomics analysis, decreases the confidence in the study.

Major points:

1) The authors conclude that the GS DIPG exhibits elevated cholesterol metabolism and OXPHOS; therefore, these cells are more vulnerable to perturbations of these pathways. However, GS cells show decreased bioenergetic capacity and a general decrease in TCA cycle intermediates, thus contradicting some of these conclusions for OXPHOS. Upregulation in gene expression could also suggest a compensatory mechanism for the decreased cholesterol metabolism or OXPHOS? It is unclear whether, indeed, there is indeed an increased flux through cholesterol metabolism and OXPHOS? Can the authors assess these pathways more directly through metabolic tracing? Are there differences in NAD/NADH ratio between GS vs. DGC state?

2) It is unclear whether the astrocyte (AC)-like cells form tumors and whether they are sensitized to ferroptosis in vivo models.

3) Several figures are missing quantification. (Fig. 1B, 3A, 3C, 3E, F, 4A-E, and others in the Supplementary materials).

Reviewer #4 (Remarks to the Author): Expert in ferroptosis and metabolic vulnerabilities

Review of manuscript NCOMMS-22-08022, "Therapeutic targeting of differentiation state-dependent metabolic vulnerabilities in DIPG", by Mbah et al., et al.

In the present study the authors have used patient derived cell lines to study the heterogeneity of H3K27M diffuse intrinsic pontine gliomas (DIPG). In brief, DIPG heterogeneity is associated to two phenotypic states, a less differentiated, called OPC-like and a more differentiated, astrocyte (AC)-like. In this study, the authors used a number of omics methodologies to characterize cells that grow as spheres (GS) or adherent in the presence of FBS (DGC). Their characterization revealed that GS present OPC-like genetic signature while DGC are more AC-like. Ultimately, their study has identified a number of state-specific dependencies that could be leveraged for therapeutic gain. In general this is an interesting work with a number of unexpected observations.

My review of the manuscript, as requested by the editorial team, focuses solely on the ferroptosis aspect of this work (Figure 3).

In this specific part of the work, the authors suggest that the AC-like (DGC) up-regulate EMT-associated genes ultimately resulting in an increased sensitivity to ferroptosis, as suggested by previous works. My major concern regarding this experiments is the following, GS cells are grown in the presence of B27 a supplement known to be rich in enzymatic (SOD and CAT) and non-enzymatic antioxidants (vitamin E analogues, selenium, GSH). I would be surprised if DGC, which are sensitive to ferroptosis, cultured in the presence of B27 show a profile different from that of GS cells treated with RSL3. Having said this, the authors do try an inverse control, which is to treat culture GS cells in AC-media but this could be misleading. Perhaps a more suitable control would be to use B27 lacking antioxidants (this is commercially available, which makes it easy to be implemented).

Up until today the role of EMT in ferroptosis is far from clear, since not always EMT is able to

sensitize to ferroptosis. Therefore, another point that could be better explored, is a better characterization of the model in terms of canonical ferroptosis regulators (GPX4, SLC7A11, AIFM2/FSP1, ACSL4 – for example). I believe this could be informative. Please keep in mind that B27 will boost enormously the expression of selenoprotein, such as GPX4.

Minor:

1- the authors use GDC and DGC in the text, that was a bit confusing. Please correct.

2- In the graphs 3C and 3D, to what "mock" refers to? To my understanding, this is superfluous; if not, a short note in the figure legend would be helpful.

Rebuttal of Mbah, et al.

We are truly grateful for the time and effort put forth by the Editors and Reviewers at *Nature Communications*. We have addressed all the experimental and textual concerns raised, and we believe that the helpful comments have allowed us to considerably strengthen the conclusions presented in the accompanying manuscript. Below is a point-to-point response to these comments; referee remarks are presented in plain text, our responses in bold.

REVIEWERS' COMMENTS

Reviewer 1

The authors of "Therapeutic targeting of differentiation state-dependent metabolic vulnerabilities in DIPG " employed a systems wide approach to characterize in vitro cellular heterogeneity of H3K27M+ using isogenic in vitro models comparing stem-like glioma cells (resembling oligodendrocyte precursors -OPC) and more differentiated astrocyte (AC)-like DIPG cells. In doing so they have identified metabolic programs operative in the different in vitro differentiation states, helping to identify treatment vulnerabilities. The potential for targeting metabolism may go some way to combat the inter- and intratumoral heterogeneity of the disease if similar influences are seen in vivo. The authors have revealed the potential of several clinically relevant metabolic drugs, important because as it stands, outcomes for patients diagnosed with DIPG remain are unacceptably poor. I have several major points that need to be addressed.

Response: We thank the reviewer for the supportive comments and suggestions. We have incorporated this feedback, and it has served well to provide clarity and further strengthen our conclusions.

Major:

The authors describe cell populations as either isogenic GS or isogenic DGC. Can the authors describe whether they in fact know they are isogenic following the change in cellular state?

Given studies confirming the intratumoral heterogeneity of patient tissues and cell lines, approximately 6 subclones are present in each tumor / cell line (PMID: 29967352; PMID: 26727948), so it is indeed possible that the change in cell culture condition helps to enrich for colonies/clones of particular genotype and cellular type?

Response: We referred to these cell populations as isogenic since the differentiated glioma cells (DGC) were directly generated from the gliomaspheres (GS) by culturing the GS in the presence of serum. This is common nomenclature in the field for cell lines genetically modified, for example, to express an oncogene. Formally, isogenic means the same genotype. Since our cells were not modified, in principle, the genotype remains identical between the differentiated DGC and the GS. In support of this, we did not observe cell death during differentiation of GS to DGC, arguing against the selection of a rare subclone, as the references provide could suggest. Further, we performed and now provide STR profiling on the paired lines, which indicates that they are genetically identical at the loci assessed (Supplementary Figure 1B).

However, we appreciate the spirit of the reviewer's suggestions. So as not to extend our conclusions beyond what we formally know, we removed the isogenic descriptor from the manuscript, referenced the papers indicated, and noted that we could not formally rule out the possibility that we selected a subclone during the differentiation.

In the PCA analysis of RNA expression data in Supp Fig. 1, the authors use PC2 vs. PC3 distinguish GS from DGC cells, suggesting that PC3 is more influenced by the cellular differentiation status than the cell type. For the most part this is true, except for SU-DIPG-13 which clusters independent of +/-serum. Remarkably, 1329 genes and 1163 genes were commonly differentially up- or down- regulated, GS vs. DGC respectively, across all three cell types. Given, serum fails to change the cellulotype or principal components of SU-DIPG-13 it is interesting that so many genes are conserved compared to cells where serum induces such a pronounced change in growth and differentiation (PCA)?

Response: We thank the Reviewer for their careful analysis of our work and the insightful inquiry on gene expression patterns. The principal component analysis (PCA) was useful for identifying molecular differences between the cells in an agnostic manner. To address the Reviewer's comment about the interesting observations with DIPG-XIII, we performed a direct comparison of DIPG-XIII relative to DIPG007 and SF7761 under adherent culture or when grown as neurospheres. We then determined the core genes that were upregulated or downregulated in DIPG-XIII irrespective of culture condition, i.e., the gene expression profile unique to DIPG-XIII. Whereas the genes upregulated in DIPG-XIII showed little enrichment in pathways related to cancer, the pathway annotation of genes downregulated in DIPG-XIII were clearly cancer-associated pathways, notable among which were 'endocytosis', 'pathways in cancer', 'p53 signaling', 'Hippo signaling' and 'oxidative phosphorylation' (Supplementary Figure 1C). These data now make clearer that DIPG-XIII cells are indeed different from the other two cell lines, which is also consistent with and could explain the distinct phenotypic observations we made with this cell line throughout the manuscript.

It is important to note that SU-DIPG-13 were established from an autopsy, of a frontal lobe metastasis, whereas the SU-DIPG-13P* were established from the pontine region of the same patient (PMID: 28434841).

Given changes in EMT encourages metastatic dissemination, and the importance of the tumor microenvironment on transcriptional selection pressures that regulate growth, metabolism, differentiation, immune recognition, the recognition that these cells are different may go way to describe the sensitivity differences shown SU-DIPG-13 vs. SU-DIPG-13P*. Do SU-DIPG-13P* cells show a visual differentiation change +/- serum?

If so, do GS established from these cells show the predictive markers of OPC/GS-like vs DGC/AC, revealed?

Response: We discovered that the SU-DIPG-13P* cells in our labs have been mislabeled and are instead SU-DIPG-13. During the revision process, we performed formal, direct comparison of morphology and response to statins in what we labeled as SU-DIPG-13P* and SU-DIPG-13 and, unsurprisingly, found no difference. We have removed all the previous data using SU-DIPG-13* from the paper. We are truly grateful for the opportunity to correct this mistake.

Can the authors provide a summary of expression differences between the cell types that may help to describe why SU-DIPG-13 shows no change in morphology or growth rate? It may be possible that these metastatic cells are more amenable to other therapies shown to work in brain metastasizes, that fail in primary brain tumors.

Response: First, we thank the referee for the opportunity to clarify the point on morphology. Marked changes are observed in the DIPG-XIII cells grown in serum. This was difficult, if not impossible, to observe with the previous Extended Data Figure 1. We now provide a higher magnification image alongside the previous image (Extended Data Figure 1A).

To address pathways that may account for the lack of change in growth rate, we performed a differential gene expression analysis for pathways unique to DIPG-XIII (Supplementary Figure 1C), detailed in Response #2 above. In short, we observe that “p53 Signaling Pathway” and “Pathways in Cancer” are uniquely down in the DIPG-XIII cells, relative to the other lines. We believe that the differential regulation of such pathways likely accounts for the lack of change in growth rate. A more detailed analysis is an excellent lead for a future study.

I am wondering what would be the effect of the addition of serum if the cells were grown in ultra-low adherent flasks?

Response: Addition of serum to DIPG-007 GS grown in non-tissue culture treated plates resulted in adherence (Supplementary Figure 1A). Removal of growth factors further potentiated the adherence of DIPG-007 GS to non-tissue culture treated plates.

The authors provide interesting stem and differentiation markers GS/DGS and list individual gene expression profiles for each of cell lines in Supplementary Fig. S2. These data highlight the conserved expression of OLIG1 and OLIG2 in GS across cell types. However, the authors have neglected to provide any statistics here.

Response: We thank the referee for pointing out this oversight. Statistics have been added for this figure in Supplementary Table 2, as well as throughout the manuscript.

PDGFRA and BIM1 expression have previously been associated with oligodendrocyte precursor cells (OPC-like) from primary tumors. The data here shows GS expression profiles also enrich for the OPC-like gene expression signature identified by single cell RNA sequencing (PMC5949869). It is interesting that differentially, SOX2 and EGFR showed increased expression in cell types that show a growth and morphology change i.e., GS vs DGS, however not in SU-DIPG-13, which remained morphologically (transcriptionally?) the same regardless of GS vs DGS status. Given cell culture media is supplemented with EGF it would be interesting to see if the changes in morphology and growth rates are affected by the availability of hu-EGF (and potentially hu-PDGF-AA and hu-PDGF-BB, given no change in PDGFRA was seen in GS cells). It is well characterized that EGFR signaling promotes SOX2 expression, which in turn binds to the EGFR promoter to directly upregulates EGFR expression, providing a plausible link describing the differences in cell type shown herein.

Response: We thank the reviewer for this insightful question. All GS cultures were propagated in EGF, FGF, and PDGF-AA/BB, and experiments with GS cultures were similarly carried out in the presence of full growth factors. Moreover, we discovered that these GS cultures lose viability if grown in the absence of EGF and FGF. Thus, despite equal amounts of EGF, expression of EGFR and SOX2 are elevated only in DIPG-007 and SF7761 GS (Extended Data Figure 2). This suggests that there are other factors contributing to EGFR expression beyond supplementing with EGF/FGF.

The clinical relevance of the GS gene expression profiles was investigated and correlated with OS data provided by a patient dataset. Given the differences in expression profiles in the SU-DIPG-13, I am interested to know if the authors were to exclude these data and mine GS high transcripts assessed just from DIPG007 and SF7761 GS cells, what would the survival outcome look like?

Response: These data have been included in Extended Data Figure 2J of this revised study, illustrating a statistically significant greater survival among patients with low GS gene signature.

Each of the cell lines used are H3.3, with the Kaplan-Meier survival prediction looks remarkably like H3.3K27M vs. H3.1K27M? Can the authors provide more statistics for Fig. 1H including median OS, CI Log Rank test results?

Response: Median survival is 11.9 months versus 8 months, and the Log Rank $p=0.0176$. These data are provided in the legend for Figure 1 in this revised study.

The glycolytic and glutamine metabolism signatures enriched in GS vs. DGC are also very interesting. As the authors highlight metabolic dependencies have been investigated in H3K27M gliomas revealing dependencies on glucose and glutamine metabolism. Did the RNA sequencing show changes in the expression of GLUT receptors or glutamine transporters in GS cells?

Response: We thank the reviewer for this insightful suggestion. We did not observe a consistent pattern of differentiation state-dependent change in SLC2A1 (GLUT1), SLC2A3 (GLUT3), or SLC1A5 (ASCT2) among the cell line pairs (Supplementary Figure 2A-C). SLC2A1 is higher in GS for SF7761 and DIPG-XIII, while significantly lower in DIPG-007; SLC2A3 is higher in GS for DIPG-007 and SF7761, while significantly lower in DIPG-XIII; and SLC1A5 is higher in GS for DIPG-007 and SF7761, no changes were observed for DIPG-XIII.

Can modulation of glucose or glutamine change morphology?

Response: To address this question, we cultured GS vs DGC in glucose or glutamine free media and assessed morphology and viability/proliferation after 7 days. For glucose, we used 0mM, 2.5mM, and 21mM, and for glutamine, we cultured cells in 0mM or 4mM. In the GS, there was a clear decrease in proliferation in the glucose-deprived conditions, while the decrease for DGC was modest (Supplementary Figure 2D). Proliferation was not

impacted for either differentiation state in the glutamine-free media (Supplementary Figure 2D). Differences in cell morphology were not observed for GS or DGC after 7 days of glucose or glutamine deprivation (Supplementary Figure 2E,F).

Fig 3C shows GPX4 inhibition to cause cell death selectively in GDC cells at low concentration of RSL3. However, SF7761 GDC and GS show a similar IC50 between to RSL3. Do these cells show transcriptional differences that might help to explain the change in sensitivity seen?

Response: In this experiment, Fer-1 is used as a ferroptosis rescue agent. Cell death induced by RSL3 and rescued by Fer-1 can be considered ferroptotic, for these purposes. Cell death induced by RSL3 that is not rescued by Fer-1 is off-target and non-ferroptotic (PMID: 31574461). The important data to extract from this experiment is that there is a window of ferroptosis that can be induced in SF7761 DGC that is not observed in the GS.

I am also interested to know if GDC cells show a more immunogenic transcriptome?

Response: We thank the reviewer for the question. Based on GSEA of the RNAseq data, the DGC cells do indeed exhibit a more immunogenic transcriptome. The inflammatory response, TNF alpha signaling, complement pathway, IL2_STAT5, IL-6_JAK6 signaling, interferon alpha, and interferon gamma signaling pathways were generally upregulated in DIPG DGC compared to GS (Extended Data Figure 3B).

Does targeting cholesterol biosynthesis or OXPHOS change the differentiation status (GS) or morphology?

Response: As now pictured in Supplementary Figure 3F, long-term treatment with low dose statin or phenformin (up to 14 days) did not alter morphology of GS cells.

Likewise, does enhancing ferroptosis drive dedifferentiation?

Response: Like the above study with statins and complex I inhibitors, propagation of GS cells at a dose of RSL3 that induces ferroptotic cell death in the cognate DGC line does not drive dedifferentiation or alter morphology of GS cell type after 14 days of culture (Supplementary Figure 3E).

Ionizing radiation experiments in Figure 5G, require statistical evaluation?

Response: Statistical analyses have now been included for the data in this figure (revised Figure 7A).

Given the mitochondrial spare respiratory capacity (SRC) result and the clinical relevance of metformin/phenformin, I am wondering about the combination index when used in combination with radiotherapy?

Response: To address this excellent suggestion, we treated DIPG-007 GS cultures with a dose response of atorvastatin or phenformin plus or minus 2 Gy radiation. At some combinations, modest additive cell killing was observed. However, synergy was not observed, and the additivity window was narrow (Supplementary Figure 4J,K). Nevertheless, we remain encouraged by the additivity results and are exploring this combination in vivo as the subject of a future study.

Further, cholesterol biosynthesis and mitochondrial SRC results are potentially very clinically relevant given the in vitro sensitivity of all cells to atorvastatin/pitavastatin, and DIPG 13P*, 007 and SF7761 to phenformin/metformin (metformin-SF7761?). CNS multiparametric optimization suggests that pitavastatin scores 4.81/6 and phenformin scores 3.57/6, highlighting the use of pitavastatin in vivo and potential of phenformin to cross the blood brain barrier, but not in high concentrations (PMID: 26991242). Do the authors hypothesize that modulation of glucose homeostasis is playing a role in the survival extension shown?

Response: This is a great point, and as the referee undoubtedly knows, a long-standing argument concerning the anti-cancer effect of biguanides (PMID:23999444, PMID:25456737). We put forth the activity of statins and complex I inhibitors in our model is cancer cell autonomous, based on the in vitro doses applied and their selectivity. Guided by the referee's insightful question, we have added a discussion of this concept in the revised paper.

In addition, as a matter of completeness, we have performed metformin dose response studies in all the cell models. These data are now presented in Figure 4A and Extended Data Figure 6B,G.

The authors indicate that (lines 404-405) "Notably, treatment with pitavastatin or phenformin resulted in the reduction in tumor volume based on BLI". Can the authors please provide more clarity to Figure 6C?

Response: We thank the referee for the opportunity to clarify this figure and legend (revised Figure 7C). In this revised manuscript, clarifying text has been added to the legend, and the axes were updated in the figure. For further information about this experiment, a detailed protocol is provided in the materials and methods.

The single agent PDX results are very encouraging, however, please include more clear statistical evidence.

Response: Exact statistical values and the test used are now provided in the figure legend, per the referee's suggestion.

Previous studies using glioma stem cells tested phenformin using both in vitro and in vivo models. Phenformin inhibited the self-renewal of glioma stem cells decreased the expression of stemness and mesenchymal markers, however at twice the concentrations (PMID: 27486821), these studies tested phenformin in PDX models using 100 mg/kg. Given the dose dependent decrease in proliferation of DIPG007 GS in vitro, and the increased cleaved PARP shown in Figure 4 (I acknowledge that at the same concentration no increase in in vitro apoptosis was

shown in glioblastoma stem cells- PMID: 27486821), I am wondering why the authors chose to use this dose in their in vivo experiments?

Response: We conducted a two-week dose escalation study and found the 50mg/kg phenformin to be suitable for long-term daily administration without signs of distress or weight loss (Extended Data Figure 9C).

Given that the majority of successful PDX models are established from neurosphere lines, DIPG007, DIPG13, SU-DIPG-13P*, it would be interesting to know whether DIPG007 cells induced to grow as GDC engraft?

Response: We thank the reviewer for this great question. We implanted DIPG-007 DGC into the pons of mice at the same cell concentration and with the same approach described in the methods section for the GS studies. Tumor growth was monitored using bioluminescent imaging up to 9 weeks post-injection. Consistent with previous reports that differentiated DIPG cells do not establish tumors in pons of mice (PMID: 29674595), we found that 6 of 7 mice showed no establishment of tumor growth using DIPG-007 DGC cells (Extended Data Figure 9A). One mouse had a small tumor with peak luminescence measured to be 501 luminescence units at week 9. This signal is at the lower extreme of the range we use to classify whether a tumor is quantifiable. In contrast, DIPG-007 GS cells formed discernable, large tumors by 7 weeks with an average peak luminescence of ~4,000 units (Extended Data Figure 9B).

Given the mice still succumb what is the differentiation status of the tumors at endpoint treatments vs. vehicle?

Response: This is an excellent inquiry. To answer this question, we stained endpoint tumors for cleaved caspase 3 (CC3) to assess apoptosis; Ki67 to assess proliferation; and GFAP, S100, and Olig2 to assess differentiation. We did not observe a statistically significant difference for any marker or any treatment (Extended Data Figure 10).

Additionally, given the data provided shows no significant change in animal weight, or BLI, with phenformin alone or in combination with pitavastatin, it would be important to know whether the treatments caused apoptosis in vivo, and if not at these concentrations, what dose can be achieved to cause DIPG specific apoptosis? Did the authors see any lactic acidosis in their animals?

Response: We appreciate the importance of this question. However, our study was not designed to assess the short-term impact of drug on tumor progression or to assess PD markers (including lactic acidosis). Tumor-bearing animals were treated from weeks 3-11, and animals died or were humanely sacrificed more than 1 month later (Figure 7B). Accordingly, the short-term impact of drug treatment is not expected in endpoint tumors. Nevertheless, we did assay proliferation (Ki67) and apoptosis (CC3), as described above, for which we did not observe statistically significant differences in treated animals (Extended Data Figure 10). We consider this an important point and are pursuing therapeutic assessment and combination studies in future studies.

Minor –

The DIPG acronym in the title should be expanded.

Response: This has been updated.

Word alignment in Figure 1A needs addressing.

Response: This has been updated.

Nomenclature for proteins listed in captions needs addressing.

Response: This has been updated throughout.

In the PCA analysis of RNA expression data in Supp Fig. 1, can the authors please inform why there are three (S1B) and four (S1C) individual results for SF7761 samples when there are only two comparisons in each PCA?

Response: For the RNA expression analysis, triplicate samples were prepared and analyzed for each of the six conditions. Outside of the samples noted above for SF7761, the rest of the samples are so similar that the data points fall on top of each other in the PCA. The SF7761 replicates, while highly similar, are sufficiently dissimilar to see each individual replicate.

Statistics from Fig. 3A are missing.

Response: Statistics have been added to the legend for Figure 3A.

Mouse weights should be presented as % change from baseline, rather than the actual weight.

Response: This has been updated in Figure 7D and Extended Data Figure 9C.

Add unit of measurement in BLI experiment Figure 6C.

Response: BLI is measured in luminescence (a.u.). Figure 7C has been updated.

The survival analysis requires the inclusion of more statistical evidence.

Response: Exact statistical values and the test used are now provided in the figure legend.

The methods lack a description of SU-DIPG-13P*.

Response: As noted above, this cell line has now been removed from the manuscript.

Reviewer 3

In this manuscript by Mbah et al., the authors sought to investigate the metabolic programs that operate in diffuse intrinsic pontine gliomas (DIPG), with the goal of identifying metabolic vulnerabilities for this deadly disease. The authors established in vitro isogenic models from three patient-derived H3K27M DIPG models in two distinct cellular states: 1) less-differentiated or stem-like gliomaspheres (GS) or 2) differentiated glioma cells (DGC).

Using transcriptomics and metabolomics in the GS versus DGC cell states, the authors find that less-differentiated cells (GC) enrich for gene and metabolic signatures related to OXPHOS and cholesterol metabolism and that these GC cells were more vulnerable to inhibition of OXPHOS or statin treatment both in vitro and in an orthotopic model of DIPG. Interestingly, the more differentiated astrocyte (AC)-like cells showed a mesenchymal phenotype and these cells displayed sensitivity to ferroptosis.

This study suggests that metabolic inhibitors of OXPHOS or statins could be used to target DIPG for therapeutics. The study is interesting and clearly presented. However, I have some reservations about how generalizable these metabolic programs are for DIPG, given that the study is based on three cell lines. Moreover, the conclusions of the metabolic programs regulated in various cellular states rely primarily on steady-state metabolomics and transcriptomics data. The lack of metabolic tracing measurements, or proteomics analysis, decreases the confidence in the study.

Response: We thank the referee for their careful reading of our work, enthusiasm, and supportive feedback while also highlighting areas to improve the strength of our conclusions. Guided by the referee's constructive criticism, we have employed additional metabolic measurements to determine metabolic activity and flux more accurately.

Major points:

1) The authors conclude that the GS DIPG exhibits elevated cholesterol metabolism and OXPHOS; therefore, these cells are more vulnerable to perturbations of these pathways. However, GS cells show decreased bioenergetic capacity and a general decrease in TCA cycle intermediates, thus contradicting some of these conclusions for OXPHOS. Upregulation in gene expression could also suggest a compensatory mechanism for the decreased cholesterol metabolism or OXPHOS? It is unclear whether, indeed, there is indeed an increased flux through cholesterol metabolism and OXPHOS? Can the authors assess these pathways more directly through metabolic tracing? Are there differences in NAD/NADH ratio between GS vs. DGC state?

Response: We appreciate the excellent suggestions from the referee and the opportunities to clarify and strengthen our conclusions. In brief, we previously illustrated that GS cultures exhibit increased pool sizes for glycolytic metabolites and modestly decreased pool sizes for TCA cycle metabolites by LC/MS (Figure 2D,E). To test metabolic activity, we used the Seahorse bioanalyzer, which demonstrated that GS had lower glycolytic rates (i.e., ECAR; Figure 6F) and lower oxygen consumption (i.e. OCR; Figure 6B).

In this revised study, we substantiated these metabolic assays by assessing the rate of glucose carbon entry into glycolysis, the TCA cycle, TCA cycling, de novo lipid

biosynthesis, and the sterol biosynthetic pathway using a kinetic ^{13}C -glucose tracing assay (Figure 5A-F; Extended Data Figure 7D-K; Supplementary Figure 4A-I). In brief, we grew GS and DGC cells in glucose-free media supplemented with 21.25mM U- ^{13}C -glucose, and collected samples at 15min, 60min, 4hr, and 24hr. These were analyzed by LC/MS-based metabolomics to measure glucose carbon enrichment as a function of time to assess flux. We found that glucose entry into glycolysis and the upper TCA cycle (i.e., citrate) was modestly slower in the GS cultures. TCA cycling and anaplerosis were also reduced in the GS cultures, compared to the DGC cultures.

Together these data collectively illustrate that GS cells have lower OXPHOS activity and glycolysis, which substantiates the observation that GS cells have a lower energy charge and ATP/ADP ratio (Figure 6D,E). The highly significant, yet modestly increased OXPHOS gene expression signature is likely, as the referee suggests, a compensatory increase (Figure 1I). Indeed, this lower OCR correlated with the increased sensitivity to inhibitors of complex I (Figure 4A-C). We have reworked the interpretation of these results to reflect that the decreased activity presents a metabolic vulnerability, where less inhibition is required to have the same growth inhibitory effect.

GS also exhibit increased expression of enzymes in cholesterol biosynthesis (Figure 1I). To determine the activity/flux of the sterol biosynthetic pathway, which makes cholesterol, we utilized the kinetic ^{13}C -glucose tracing data described above. We observed that glucose carbon entry into the lower TCA cycle is markedly slower in the GS cultures (Extended Data Figure 7H-J), potentially because citrate is being siphoned to support sterol and lipid biosynthesis (Figure 5B). Indeed, we observed that sterol biosynthesis is faster in the GS cultures, as measured by labeling of mevalonate (Figure 5E). These observations reflect a higher demand for GS cells on endogenously produced cholesterol, whose inhibition presents a metabolic vulnerability.

Lastly, we determined the NAD/NADH ratio by LC/MS in the three culture pairs (Extended Data Figure 8I). Significant differences were not observed for any of the GS and DGC pairs.

2) It is unclear whether the astrocyte (AC)-like cells form tumors and whether they are sensitized to ferroptosis in vivo models.

Response: We thank the reviewer for the opportunity to experimentally address this point, which was also raised by Referee #1. We implanted DIPG-007 DGC into the pons of mice at the same cell concentration and with the same approach described in the methods section for the GS studies. Tumor growth was monitored using bioluminescent imaging up to 9 weeks post-injection. Consistent with previous reports that differentiated DIPG cells do not establish tumors in pons of mice (PMID: 29674595), we found that 6 of 7 mice showed no establishment of tumor growth using DIPG-007 DGC cells (Extended Data Figure 9A). One mouse had a small tumor with peak luminescence measured to be 501 luminescence units at week 9. This signal is at the lower extreme of the range we use to classify whether a tumor is quantifiable. In contrast, DIPG-007 GS cells formed discernable, large tumors by 7 weeks with an average peak luminescence of ~4,000 units (Extended Data Figure 9B).

3) Several figures are missing quantification. (Fig. 1B, 3A, 3C, 3E, F, 4A-E, and others in the Supplementary materials).

Response: We have updated this revised manuscript to included statistics for all the data presented.

Reviewer 4

Review of manuscript NCOMMS-22-08022, " Therapeutic targeting of differentiation state-dependent metabolic vulnerabilities in DIPG", by Mbah et al., et al.

In the present study the authors have used patient derived cell lines to study the heterogeneity of H3K27M diffuse intrinsic pontine gliomas (DIPG). In brief, DIPG heterogeneity is associated to two phenotypic states, a less differentiated, called OPC-like and a more differentiate, astrocyte (AC)-like. In this study, the authors used a number of omics methodologies to characterize cells that grow as spheres (GS) or adherent in the presence of FBS (DGC). Their characterization revealed that GS present OPC-like genetic signature while DGC are more AC-like. Ultimately, their study has identified a number of state-specific dependencies that could be leveraged for therapeutic gain. In general this is an interesting work with a number of unexpected observations.

Response: We thank the referee for their time and support of our work.

My review of the manuscript, as requested by the editorial team, focuses solely on the ferroptosis aspect of this work (Figure 3).

In this specific part of the work, the authors suggest that the AC-like (DGC) up-regulate EMT-associated genes ultimately resulting in an increased sensitivity to ferroptosis, as suggested by previous works. My major concern regarding this experiments is the following, GS cells are grown in the presence of B27 a supplement known to be rich in enzymatic (SOD and CAT) and non-enzymatic antioxidants (vitamin E analogues, selenium, GSH). I would be surprised if GDC, which are sensitive to ferroptosis, cultured in the presence of B27 show a profile different from that of GS cells treated with RSL3.

Having said this, the author do try an inverse control, which is to treat culture GS cells in AC-media but this could be misleading. Perhaps a more suitable control would be to use B27 lacking antioxidants (this is commercially available, which makes it easy to be implemented).

Response: We thank the referee for their careful reading of our work. We recognized this caveat at the early stages of our study and conducted the ferroptosis viability studies for DIPG-007 and DIPG-XIII using commercially available B27 without the antioxidant cocktail. We apologize for omitting this critical detail in the methods section of our initial publication. This has been rectified in the revised study.

As it relates to the data, we found that in a side-by-side comparison, DIPG-007 and DIPG-XIII GS cells remain equally resistant to ferroptosis with or without the antioxidants in B27 supplement (Supplementary Figure 3A-C). SF7761 cells cultured with the antioxidant-free B27 grew poorly and exhibited a massive reduction in viability. We appreciate the opportunity to clarify this important point and to accurately update our revised manuscript.

Up until today the role of EMT in ferroptosis is far from clear, since not always EMT is able to sensitize to ferroptosis. Therefore, another point that could be better explored, is a better characterization of the model in terms of canonical ferroptosis regulators (GPX4, SLC7A11, AIFM2/FSP1, ACSL4 – for example). I believe this could be informative.

Response: We appreciate this excellent suggestion. Data for the above noted genes are now presented in Supplementary Figure 3D. In short, we did not observe consistent changes for these canonical ferroptosis regulators for a given differentiation state across our cell line panel.

Please keep in mind that B27 will boost enormously the expression of selenoprotein, such as GPX4.

Response: As the referee notes, B27, by way of Selenium (Se), in principle can promote GPX4 expression and thus resistance to ferroptosis. Consistent with this, we observed a significant, albeit modest, increase in GPX4 expression in DIPG-007 and SF7761 GS cultures (grown in B27) relative to DGC cultures (Supplementary Figure 3D). This was not observed in DIPG-XIII. However, as described above, GS cells remain resistant to ferroptosis in the absence of B27 (Supplementary Figure 3A-C).

Minor:

1- the authors use GDC and DGC in the text, that was a bit confusing. Please correct.

Response: We apologize for this oversight, which has been corrected in the revised manuscript.

2- In the graphs 3C and 3D, to what “mock” refers to? To my understanding, this is superfluous; if not, a short note in the figure legend would be helpful.

Response: Thank you, we have updated the figures and figure legend replacing “mock” with “control”.

REVIEWER COMMENTS

Reviewer #1 (Remarks to the Author):

I would like to express my gratitude for the courteous responses to my review of this paper. The authors have diligently addressed my points, resulting in significant improvement to the manuscript.

I would like to draw attention to my previous comment regarding the subclonal architecture of DIPG and the change in cell state, influenced by varying culture conditions. It seems there is a discrepancy in the authors' statement in their rebuttal letter, where they mention that differentiation did not affect cell viability, while in the revised text (line 127), they acknowledge observing cell death during differentiation from GS to DGS. Additionally, I find the STR profiling results surprising, indicating genotype change. We see that biopsy to established culture selects different genotypes, here however, the selection of cell subclones may no longer represent the original biopsy due to the number of passages of these cell lines.

I must note the incomplete statistical analysis throughout the revised manuscript. For comparative growth and proliferation analysis, I suggest employing the area under the curve (AUC) for assessing differences between treatments and growth stimuli, not only in main results but also in supplementary and extended figures.

It's regrettable that a cell mix-up occurred. As previously mentioned, DIPG13 is likely reflective of its microenvironment at autopsy being the metastasis of DIPG13P*. Given the limited differences observed in growth rates, transcriptional changes, and similar responses to OXPPOS and statins between GS and DGC, a comparison would have been insightful. Furthermore, despite similarities in growth, proliferation, and transcriptional responses to serum and OXPPOS inhibitors between GS and DGC, the striking difference in morphology is striking?

Could the authors address my previous question regarding the high vs low survival analysis of GS? Overlaying patient overall survival (OS) with H3.3 vs H3.1 or conducting independent analyses may elucidate the role of canonical vs non-canonical histones in the differentiation state.

The carbon labeling experiments lack statistical comparison. From a broad perspective, the accumulation of labeled carbons seems to correspond more with time than differentiation state. There are font issues in Figure 5K.

Overall, the paper has made significant strides, but these points require further attention.

Reviewer #3 (Remarks to the Author):

The authors have addressed my comments.

Reviewer #4 (Remarks to the Author):

The authors have addressed my points. Thank you for carefully considering them.

I only have a minor point if the authors want to discuss it. The authors identified that GS cells are more resistant to ferroptosis and showed an increase in the activity of the mevalonate pathway in this cell line. Is there any connection between products of the mevalonate pathway, such as squalene and 7-dehydrocholesterol, that are responsible for this increased sensitivity? It's probably not necessary to validate this, but I think it could enrich the discussion.

Second Rebuttal of Mbah, et al.

In this second rebuttal, we have addressed the experimental and textual concerns raised. Below is the point-to-point response to these comments; referee remarks are presented in plain text, our responses in bold.

REVIEWERS' COMMENTS

Reviewer 1

I would like to express my gratitude for the courteous responses to my review of this paper. The authors have diligently addressed my points, resulting in significant improvement to the manuscript.

I would like to draw attention to my previous comment regarding the subclonal architecture of DIPG and the change in cell state, influenced by varying culture conditions. It seems there is a discrepancy in the authors' statement in their rebuttal letter, where they mention that differentiation did not affect cell viability, while in the revised text (line 127), they acknowledge observing cell death during differentiation from GS to DGS.

Response: We thank the reviewer for their careful reading of our work. There was a typo in the manuscript in the line referenced above. This has been updated, as follows:

“During the differentiation procedure, we did **not** observe cell death, arguing against the selection of a rare subclone^{34,35}.”

Additionally, I find the STR profiling results surprising, indicating genotype change. We see that biopsy to established culture selects different genotypes, here however, the selection of cell subclones may no longer represent the original biopsy due to the number of passages of these cell lines.

Response: The STR profiling results, presented in Supplementary Figure 1B, illustrate that the genotypes of the DGC and GS cultures are identical between each of the three cell line pairs. In other words, we did not observe a genotype change between DGC and GS within any of the cell line pairs.

I must note the incomplete statistical analysis throughout the revised manuscript.

Response: We again thank the referee for their careful attention to detail. In this revised manuscript, a more thorough and detailed statistical analysis is provided. This information is provided in the figures and figure legends, and/or, when space is limited, full statistical values are presented in updated Supplementary Table 2.

For comparative growth and proliferation analysis, I suggest employing the area under the curve (AUC) for assessing differences between treatments and growth stimuli, not only in main results but also in supplementary and extended figures.

Response: We appreciate this excellent suggestion for assessing statistical significance between treatments and growth stimuli. AUC has now been employed for all statistical comparisons in this revised study.

It's regrettable that a cell mix-up occurred. As previously mentioned, DIPG13 is likely reflective of its microenvironment at autopsy being the metastasis of DIPG13P*. Given the limited differences observed in growth rates, transcriptional changes, and similar responses to OXPPOS and statins between GS and DGC, a comparison would have been insightful. Furthermore, despite similarities in growth, proliferation, and transcriptional responses to serum and OXPPOS inhibitors between GS and DGC, the striking difference in morphology is striking? Could the authors address my previous question regarding the high vs low survival analysis of GS? Overlaying patient overall survival (OS) with H3.3 vs H3.1 or conducting independent analyses may elucidate the role of canonical vs non-canonical histones in the differentiation state.

Response: A comparison of OS using the GS signatures for patients with H3.1 mutations is now presented in Extended Data Figure 2K alongside the previous data for patients with H3.3 mutations. That said, while the p value is significant, the direction of the curve flips in H3.1K27M vs H3.3K27M tumors, though the n is low for H3.1K27M (total of 12, n=4 vs n=8 for quantiles), precluding definitive conclusions.

The carbon labeling experiments lack statistical comparison. From a broad perspective, the accumulation of labeled carbons seems to correspond more with time than differentiation state.

Response: Statistics have now been included. The accumulation of label as a function of time is expected; as cells metabolize glucose, label accumulates. The differences in pool size between GS and DGC are statistically significant, and the magnitudes of difference are in line with what we and others have seen when comparing fractional enrichment patterns between cell types. For example: pancreatic cancer cells +/- mutant Kras (PMID: 22541435, PMID: 25119024) or pancreatic cancer cells +/- GOT1 expression (PMID: 31908776).

There are font issues in Figure 5K.

Response: Figure 5K has been closely scrutinized for accuracy and clarity in this revised manuscript.

Overall, the paper has made significant strides, but these points require further attention.

Response: We again thank the reviewer for their careful attention to detail, supportive comments, and suggestions. These additions have served well to provide needed transparency and clarity.

Reviewer #3 (Remarks to the Author):

The authors have addressed my comments.

Response: We thank the referee for their time and support.

Reviewer #4 (Remarks to the Author):

The authors have addressed my points. Thank you for carefully considering them.

I only have a minor point if the authors want to discuss it. The authors identified that GS cells are more resistant to ferroptosis and showed an increase in the activity of the mevalonate pathway in this cell line. Is there any connection between products of the mevalonate pathway, such as squalene and 7-dehydrocholesterol, that are responsible for this increased sensitivity? It's probably not necessary to validate this, but I think it could enrich the discussion.

Response: Thank you. We appreciate this excellent suggestion. While we did not measure squalene or 7-dehydrocholesterol in these assays, we recognize the potential insight and have included mention of putative role(s) in the revised discussion.

REVIEWERS' COMMENTS

Reviewer #1 (Remarks to the Author):

I thank the authors for addressing my comments with care and diligence. I have no further questions.